# Learning on a Razor's Edge: Identifiability and Singularity of Polynomial Neural Networks

**Vahid Shahverdi \*, Giovanni Luca Marchetti \* & Kathlén Kohn \***
Department of Mathematics
KTH Royal Institute of Technology
Stockholm, Sweden
`{vahidsha,glma,kathlen}@kth.se`

## Abstract

We study function spaces parametrized by neural networks, referred to as neuromanifolds. Specifically, we focus on deep Multi-Layer Perceptrons (MLPs) and Convolutional Neural Networks (CNNs) with an activation function that is a sufficiently generic polynomial. First, we address the identifiability problem, showing that, for almost all functions in the neuromanifold of an MLP, there exist only finitely many parameter choices yielding that function. For CNNs, the parametrization is generically one-to-one. As a consequence, we compute the dimension of the neuromanifold. Second, we describe singular points of neuromanifolds. We characterize singularities completely for CNNs, and partially for MLPs. In both cases, they arise from sparse subnetworks. For MLPs, we prove that these singularities often correspond to critical points of the mean-squared error loss, which does not hold for CNNs. This provides a geometric explanation of the sparsity bias of MLPs. All of our results leverage tools from algebraic geometry.

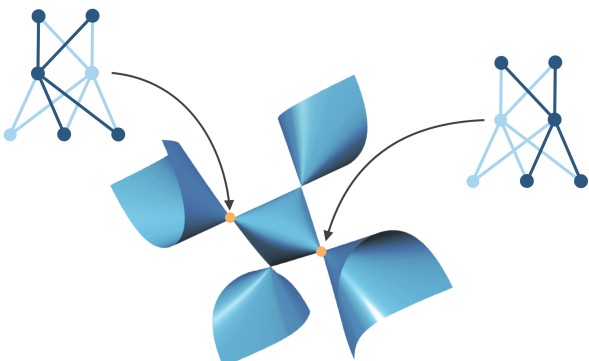

Figure 1: Subnetworks define singular points (orange) of the neuromanifold.

# 1 Introduction

Machine learning models parametrize spaces of functions, often referred to as *neuromanifolds* (Calin, 2020; Marchetti et al., 2025). Their geometry governs several learning aspects, ranging from expressivity and sample complexity to the training dynamics. Neuromanifolds are central to statistical learning theory—where they are referred to as hypothesis spaces—and information geometry (Amari, 2016), and have been analyzed from the perspective of various mathematical disciplines.

---

\*Equal contribution.

A classical question in machine learning is *identifiability*—the problem of describing the parameters that correspond to the same function. Several works have formally addressed this question for specific activation functions, such as Tanh (Fefferman et al., 1994), Sigmoid (Vlačić & Bölcskei, 2022), ReLU (Grigsby et al., 2023; Bona-Pellissier et al., 2023), and monomials (Finkel et al., 2024; Usevich et al., 2025). Identifiability can be interpreted as characterizing the redundancies or 'symmetries' of the parametrization of the neuromanifold (Lim et al., 2024). The degree of redundancy determines the *dimension* of the neuromanifold, which, in turn, measures the sample complexity of the corresponding model (Cucker & Smale, 2002). Moreover, non-identifiable functions often correspond to *singularities* of the neuromanifold. These are special points where the model is not a smooth manifold, such as edges, cusps, or self-intersections. As advocated by the field of Singular Learning Theory (Watanabe, 2009), singular points play an important role for learning, e.g., in terms of generalization error (Wei et al., 2022) and training stability (Amari et al., 2006; Wei et al., 2008).

In this work, we investigate the identifiability and singularities of both Multi-Layer Perceptrons (MLPs) and Convolutional Neural Networks (CNNs). Instead of focusing on a single activation function, we consider a large class of activation functions, namely, generic polynomials of large degree. This is motivated by the fact that polynomials can approximate arbitrary continuous functions, and thus, polynomial neural networks approximate arbitrary networks. Crucially, considering polynomial activations makes geometric analysis more feasible for two reasons. First, polynomial MLPs are the only ones whose neuromanifolds live in finite-dimensional ambient spaces (Leshno et al., 1993, Theorem 1). Second, it allows us to employ tools from algebraic geometry (Kileel et al., 2019; Trager et al., 2020). The latter is a rich field of mathematics that is particularly suitable for analyzing symmetries and singular spaces. Polynomial networks are the central focus of *neuroalgebraic geometry* (Marchetti et al., 2025)—an emerging field bridging algebraic geometry and theoretical deep learning, where our work naturally fits.

Our main results apply to sufficiently generic polynomial activation functions of large degree:

**Identifiability:** We prove that almost all functions in the neuromanifold of an MLP correspond to only finitely many parameters (Theorem 4.1). This result is in line with the widespread belief that general MLPs have discrete parameter symmetries (coming from permutations of neurons in each layer). However, the only available proofs of this claim are for Tanh and Sigmoid activations (Fefferman et al., 1994; Vlačić & Bölcskei, 2022). Our result extends the identifiability studies for monomial activations (Finkel et al., 2024; Usevich et al., 2025) to the broad class of generic polynomials. In particular, we deduce that the dimension of the neuromanifold coincides with the number of parameters. This resolves the dimension conjecture by (Kileel et al., 2019) in a general form. For CNNs, we prove a stronger result: almost all functions are uniquely identifiable, i.e., they correspond to a single choice of parameters (Theorem 4.4).

**Singularities:** We consider sparse *subnetworks*, where a subset of neurons is inactive. We prove that these subnetworks (under appropriate assumptions) are singular points of the neuromanifold, both for MLPs (Theorem 4.2) and CNNs (Theorem 4.6). We then introduce the notion of *critically exposed* parameter sets, that contain critical points of the mean-squared error loss with positive probability over the dataset (Definition 5). We prove that subnetworks of MLPs are critically exposed (Theorem 4.3), while they are not in the case of CNNs (Proposition 4.5).

Our results can be interpreted from the perspective of *sparsity bias*—the empirically-observed phenomenon that neural networks tend to discard neurons during their training process, effectively converging to a function parametrized by a smaller subarchitecture. This is closely related to the popular 'lottery ticket hypothesis' (Frankle & Carbin, 2019)—the suggestion that the model implicitly identifies and amplifies subnetworks that are particularly efficient for the given task, resulting in sparse representations of data. Our singularity results provide a geometric explanation of these phenomena. Indeed, our notion of critically exposedness formalizes the idea of a bias of the training dynamics towards the given set, since critical points are local attractors for stochastic gradient flows (Chen et al., 2023). Our results on critical exposedness of subnetworks indicate that the sparsity bias is exhibited by MLPs, but not by pure single-channel CNNs. This is in line with recent empirical findings for convolutional architectures (Blumenfeld et al., 2020).

## 2 RELATED WORK

**Algebraic geometry of deep learning.** As anticipated, a line of research in theoretical deep learning—recently termed 'neuroalgebraic geometry' (Marchetti et al., 2025)—explores the study of neuromanifolds of polynomial neural networks through the lens of algebraic geometry. Motivated by the fact that polynomials can approximate arbitrary continuous functions, the goal of neuroalgebraic geometry is to approach fundamental problems in machine learning via tools from algebra. One such problem is identifiability, which represents a core focus of neuroalgebraic geometry. Several models have been analyzed in the literature, ranging from MLPs with linear (Trager et al., 2020) and monomial activations (Kileel et al., 2019; Finkel et al., 2024; Kubjas et al., 2024; Marchetti et al., 2024), to CNNs with linear (Kohn et al., 2022; 2024; Shahverdi, 2024) and monomial activations (Shahverdi et al., 2025), to (un-normalized) attention-based networks (Henry et al., 2025). In this work, we contribute to this line of research by addressing identifiability for MLPs and CNNs with (generic) polynomial activation functions. Singularities are also central in neuroalgebraic geometry, due to their role in the learning process. However, they are formally understood only for linear MLPs (Trager et al., 2020) and monomial CNNs (Shahverdi et al., 2025). In both cases, they coincide with subnetworks of the corresponding architecture. The general relation between singularities and subnetworks is conjectural (Marchetti et al., 2025). In this work, we show that subnetworks often define singular points for MLPs and CNNs, further sedimenting this relation. 4.4.

**Implicit sparsity bias.** Several works have theoretically analyzed the tendency of neural networks to converge to sparse data representations. A line of research (Woodworth et al., 2020; Nacson et al., 2022; Pesme et al., 2021; Andriushchenko et al., 2023) has shown that, for deep diagonal linear networks, the training dynamics, when initialized around the origin, implicitly penalizes the $\ell_1$ norm of the weights, inducing sparsity in the representation. A similar bias towards low-rank solutions has been shown for deep linear networks trained with a regularized loss (Kunin et al., 2019; Ziyin et al., 2022; Wang & Jacot, 2024). Closely related to our work, these biases have been recently reformulated in terms of subnetworks (Chen et al., 2023). Most of the previous literature focuses on simple models, and on tools and ideas from (stochastic) dynamical systems theory. In contrast, we promote a purely geometric perspective, explaining the subnetwork bias in terms of the singularities and the parametrization of the neuromanifold. Moreover, we focus on general models–specifically, deep networks with polynomial activations. This generality is enabled by the powerful tools from algebraic geometry.

**Singular learning theory.** The role of singularities in machine learning—and especially their effect on the training dynamics (Amari et al., 2006; Wei et al., 2008)—has been promoted by *Singular Learning Theory* (SLT) (Watanabe, 2009; 2007; Amari et al., 2003; Wei et al., 2022). The latter falls into the more general framework of information geometry, focusing on the Riemannian geometry of the Fisher information metric. Crucially, however, the notion of singularity in SLT drastically differs from the one adopted in this work. Following the formalism of neuroalgebraic geometry, we consider singular points of the neuromanifold in the classical algebro-geometric sense—see Section 4.2 and Appendix A. In contrast, singularities in SLT are parameters where the metric tensor, once pulled back, is degenerate. This happens due to extra degrees of freedom in parameter space, or due to criticalities of the parametrization. Both these cases can lead to singular or smooth points of the neuromanifold. Vice versa, a singular point can arise in neuromanifolds that are parametrized regularly, where the pulled-back metric tensor is non-degenerate; this is the case for polynomial CNNs, as we will discuss in Section 4.4. Therefore, the two notions of singularity are different and incomparable. In a sense, our results extend ideas similar to the ones from SLT to a function space perspective, i.e., in terms of the geometry of the neuromanifold.

## 3 BACKGROUND

In this section, we overview the basic notions around algebraic geometry, deep neural networks and their neuromanifolds.

### 3.1 ZARISKI TOPOLOGY

Throughout this work, we will leverage on methods from (polynomial) algebra and algebraic geometry. Specifically, we will often use the Zariski topology, which we recall here (in the context of real vector spaces).

**Definition 1.** Fix a positive integer $n$. The *Zariski topology* on $\mathbb{R}^n$ is the topology whose closed sets are the algebraic ones, i.e., solution sets of systems of polynomial equations in $n$ variables.

Now, the Zariski topology exhibits a striking property: it is *irreducible*, i.e., any non-empty open set is dense. Moreover, dense sets in the Zariski topology are also dense in the Euclidean one. As such, for a Zariski-open set, it suffices to find a single element in it, to conclude that 'almost all' points of $\mathbb{R}^n$ belong to it. Based on this, elements belonging to open sets are referred to as *generic*. We will leverage on irreducibility in several arguments, which will allow us to obtain results that hold for generic objects (e.g., generic parameters, and generic activation functions).

### 3.2 NEUROMANIFOLDS AND SUBNETWORKS OF MLPS

Fix a function $\sigma \colon \mathbb{R} \to \mathbb{R}$, a sequence of $L > 1$ positive integers $d_0, \ldots, d_L$, and, for every $i = 1, \ldots, L$, a matrix $W_i \in \mathbb{R}^{d_i \times d_{i-1}}$.

**Definition 2.** A *Multi-Layer Perceptron* (MLP) with architecture $\mathbf{d} = (d_0, \ldots, d_L)$, activation function $\sigma$ and weights $\mathbf{W} = (W_1, \ldots, W_L)$ is the map $f_{\mathbf{W}} \colon \mathbb{R}^{d_0} \to \mathbb{R}^{d_L}$ given by the composition:

$$f_{\mathbf{W}} = W_L \circ \sigma \circ \cdots \circ \sigma \circ W_1, \tag{1}$$

where $\sigma$ is applied coordinate-wise.

We now introduce the function spaces parametrized by neural networks. Let $\mathcal{W} = \bigoplus_{i=1}^{L} \mathbb{R}^{d_i \times d_{i-1}}$ be the parameter space of an MLP, and $\varphi \colon \mathcal{W} \ni \mathbf{W} \mapsto f_{\mathbf{W}}$ be its parametrization map.

**Definition 3.** The *neuromanifold* of an MLP with architecture $\mathbf{d}$ and activation function $\sigma$ is the image of the parametrization $\varphi$, i.e.,

$$\mathcal{M}_{\mathbf{d},\sigma} = \{f_{\mathbf{W}} \mid \mathbf{W} \in \mathcal{W}\}, \tag{2}$$

As anticipated, from now on, we assume that $\sigma$ is a (univariate) polynomial of some degree $r > 1$:

$$\sigma(x) = \sum_{i=0}^{r} a_i x^i. \tag{3}$$

In this case, $f_{\mathbf{W}}$ is a (multivariate and vector-valued) polynomial of degree at most $r^{L-1}$, i.e., $\mathcal{M}_{\mathbf{d},\sigma} \subseteq \mathcal{V}$, where $\mathcal{V}$ is the space of all polynomial maps $\mathbb{R}^{d_0} \to \mathbb{R}^{d_{L-1}}$ of degree at most $r^{L-1}$. In particular, the neuromanifold lives in an ambient space $\mathcal{V}$ of functions, which is finite-dimensional and linear. This is unique to polynomial activations (Leshno et al., 1993, Theorem 1). Moreover, it follows immediately from the Tarski-Seidenberg theorem that $\mathcal{M}_{\mathbf{d},\sigma}$ is a *semi-algebraic variety*, i.e., it can be defined by polynomial equalities and inequalities in $\mathcal{V}$. The inequalities can be removed by taking the closure of $\mathcal{M}_{\mathbf{d},\sigma}$ in the Zariski topology of $\mathcal{V}$.

We also introduce the notion of a subnetwork of an MLP, which will be central in our results. For every $i = 1, \ldots, L-1$, let $A_i \subseteq \{1, \ldots, d_i\}$ and define $\mathbf{A} = (A_1, A_2, \ldots, A_{L-1})$.

**Definition 4.** A parameter $\mathbf{W} \in \mathcal{W}$ is an $\mathbf{A}$-*subnetwork* if for every $i = 1, \ldots, L-1$ and every $j \in A_i$, the $j$-th column of $W_{i+1}$ vanishes. We say that $\mathbf{W}$ is a *strict* subnetwork if, moreover, the $j$-th row of $W_i$ vanishes for $j \in A_i$.

For simplicity, we set $A_0 = A_L = \emptyset$, i.e., we do not allow subnetworks obtained by removing input or output neurons.

### 3.3 OPTIMIZATION

We consider a regression problem with mean squared error objective. To this end, let $\mathcal{D}$ be a dataset, i.e., a finite subset $\mathcal{D} \subset \mathbb{R}^{d_0} \times \mathbb{R}^{d_L}$ representing input-output pairs. The corresponding

loss $\mathcal{L}_{\mathcal{D}} : \mathcal{V} \to \mathbb{R}_{\geq 0}$ is given by:

$$\mathcal{L}_{\mathcal{D}}(f) = \sum_{(x,y) \in \mathcal{D}} \|f(x) - y\|^2. \tag{4}$$

An MLP is trained on the dataset $\mathcal{D}$ by minimizing $\mathcal{L}_{\mathcal{D}} \circ \varphi$ over $\mathcal{W}$. Since the loss is quadratic, equation 4 can be rephrased as $\mathcal{L}_{\mathcal{D}}(f) = Q(f - u)$, where $Q$ is a quadratic form on $\mathcal{V}$ and $u \in \mathcal{V}$, both depending on $\mathcal{D}$. We refer to Trager et al. (2020); Shahverdi et al. (2025); Kubjas et al. (2024) for extended discussions around this, including precise expressions for $Q$ and $u$ (depending on $\mathcal{D}$). These works also show that for large and generic $\mathcal{D}$, the quadric $Q$ is non-degenerate and $u$ is generic in $\mathcal{V}$. Therefore, with enough data, training the network amounts to optimizing a non-degenerate distance from a generic point in the ambient space of the neuromanifold.

Since optimization is typically performed via gradient descent, we will be interested in the *critical points* (in parameter space) of the loss, where its gradient vanishes. These correspond to the equilibria of the gradient flow, and are therefore the stationary points of gradient descent. For a loss $\mathcal{L}_u(f) := Q(f - u)$ as above, it follows from the chain rule that $\mathbf{W} \in \mathcal{W}$ is a critical point of $\mathcal{L}_u \circ \varphi$ if, and only if, $f_{\mathbf{W}} - u$ is orthogonal according to the scalar product induced by $Q$ to the image of the differential of $\varphi$ based at $\mathbf{W}$. This geometric interpretation of criticality will be crucial in some of our arguments.

Note that we focus on whether a point is critical, disregarding the type of criticality, i.e., (local) minima/maxima and saddles. While this enables a purely-geometric treatment, studying the type of criticality is an interesting problem, since local minima are the actual (local) attractors of the gradient flow. However, this is more subtle, and goes beyond the scope of this work – see Section 5. Moreover, it is worth mentioning that when noise is taken into account in the dynamics, arbitrary critical points can become (local) attractors, in a probabilistic sense. This is a general principle from stochastic processes, and has been discussed in the context of deep learning by Chen et al. (2023).

## 4    RESULTS

In this section, we present our main results. We first focus on MLPs, and then consider CNNs.

### 4.1    IDENTIFIABILITY OF MLPS

We now discuss identifiability of MLPs which, from the perspective of algebraic geometry, means to investigate the *fibers* of the parametrization map $\varphi$. Formally, the fiber of $\varphi$ at $\mathbf{W}$ is defined as $\varphi^{-1}(f_{\mathbf{W}}) = \{\mathbf{W}' \in \mathcal{W} \mid f_{\mathbf{W}} = f_{\mathbf{W}'}\}$.

To begin with, observe that if $\mathbf{W}$ is an $\mathbf{A}$-subnetwork, then $f_{\mathbf{W}}$ coincides with the function obtained by removing from the architecture the neurons of the $i$-th layer with indices in $A_i$, resulting in an MLP with architecture $(d_0 - |A_0|, \dots, d_L - |A_L|)$. This implies that $f_{\mathbf{W}}$ is independent of the $d_{i-1}$ entries of the $j$-th row of $W_i$ for $j \in A_i$. In particular, strict subnetworks parametrize the same function as the corresponding non-strict ones. Therefore, the fiber of $\varphi$ has a large fiber at $\mathbf{W}$—a fact has been observed in the SLT literature (Amari et al., 2006; Wei et al., 2008; Orhan & Pitkow, 2018). More precisely, this fiber has dimension $\dim(\varphi^{-1}(f_{\mathbf{W}})) \geq \sum_{i=1}^{L-1} |A_i| \, d_{i-1}$.

In our main result, we prove that, if the activation $\sigma$ is generic and of large-enough degree, the fiber of $\varphi$ at a generic $\mathbf{W} \in \mathcal{W}$ is instead finite, i.e., 0-dimensional. In other words, we show generic finite identifiability. This is equivalent, by the fiber-dimension theorem (Shafarevich, 2013, Chap. 1.6.3), to that the dimension of the neuromanifold is equal to the number of parameters. The dimension of the neuromanifold is a fundamental invariant measuring the expressivity and sample complexity of the model (Kileel et al., 2019; Marchetti et al., 2025). From a mathematical perspective, computing the dimension is an involved problem. For polynomial MLPs, it was originally conjectured in (Kileel et al., 2019), and only recently established in (Finkel et al., 2024; Usevich et al., 2025) for monomial activations $\sigma(x) = x^r$. We extend those result to more general polynomial activations. Note that monomial MLPs have infinite generic fibers, arising from neuron-wise scalings.

**Theorem 4.1.** *Suppose that $d_i > 1$ for $i = 0, \dots, L-1$, and let $\sigma$ be a generic polynomial of large enough degree $r \gg 0$ (depending on $\mathbf{d}$). Then, the generic fiber of the parametrization map $\varphi$ is*

*finite. In particular,*

$$\dim(\mathcal{M}_{\mathbf{d},\sigma}) = \dim(\mathcal{W}) = \sum_{i=1}^{L} d_i d_{i-1}. \tag{5}$$

*Proof.* Since the result is highly technical, we summarize the proof here. A full proof is provided in Appendix B.

We first show that it suffices to prove the result for a single activation. From the properties of the Zariski topology, it then follows that the result holds for a generic one. We then pick a particular activation of large degree $r$ whose coefficients are extremely sparse, i.e., the $a_i$ in equation 3 vanishes for many $0 \le i \le r$. For this activation, we argue that some homogeneous components of $f_{\mathbf{W}}$ (seen as a multivariate polynomial in its input) coincide with MLPs with the same weights $\mathbf{W}$, but where the activation function is a monomial. This implies that the fiber of $\varphi$ at $\mathbf{W}$ is contained in the intersection of the fibers of MLPs with monomial activation. As mentioned above, the generic fibers of these networks are known; they coincide with permutations of neurons at each layer, and neuron-wise rescalings. In order to conclude, we show that the intersection of the fibers of the monomial MLPs induces equations that the rescalings must satisfy. Via toric geometry we prove that those equations imply that there are only finitely many possible rescalings, concluding the proof. □

Note that an explicit bound on the degree $r$ can be extracted from the proof. Namely, by combining Lemma B.1 and Theorem B.3, it suffices that $r > (6m)^{2(L-1)^{L-1}}$, where $m = 2\max\{d_1, \ldots, d_{L-1}\}$. This bound is large, and likely to be extremely loose. Still, the existence of any bound suffices for polynomial approximation purposes, since it is possible to approximate (continuous) functions with polynomials of arbitrary-large degree – see Remark 4.1.

## 4.2 SINGULARITIES OF MLPS

Here, we show that subnetworks may yield singularities of the neuromanifold. To this end, recall that a point in a variety is singular if its tangent space has exceeding dimension, i.e., larger than the dimension of the variety; see Section A for a formal definition. Intuitively, singularities are special points where a variety exhibits degeneracy—see Figure 2 for an illustration.

Before stating our result, we discuss a simple yet illustrative case, where subnetworks are known to coincide with singularities (Trager et al., 2020; Marchetti et al., 2025). Consider a linear MLP, meaning that $\sigma(x) = x$ is the identity polynomial. Suppose that the architecture exhibits a 'bottleneck', i.e., $d_0, d_L > d := \min_{i=1,\ldots,L-1} d_i$. Then the neuromanifold contains all the linear maps $\mathbb{R}^{d_0} \to \mathbb{R}^{d_L}$ of rank at most $d$—a space known as *determinantal variety*. The geometry of the latter is well understood: the singular points coincide exactly with the maps of rank strictly less than $d$. These can be represented as subnetworks, for example, by choosing $A_i$, where $d_i = d$, of an appropriate cardinality.

We now state the main result of this section, establishing singularity of subnetworks, under an architectural assumption similar to the bottleneck from the example above.

**Theorem 4.2.** *Suppose that $\sigma$ has more than $\dim(\mathcal{M}_{\mathbf{d},\sigma})$ non-vanishing coefficients, i.e., $a_t \ne 0$ for at least $\dim(\mathcal{M}_{\mathbf{d},\sigma}) + 1$ indices $t$.[1] For every $i = 1, \ldots, L-1$, fix proper subsets $A_i \subset \{1, \ldots, d_i\}$. Suppose that for some $i = 0, \ldots, L-2$, $A_{i+1} \ne \emptyset$ and $d_i - |A_i| \le d_j - |A_j|$ for all $j \le i$. If $\mathbf{W} \in \mathcal{W}$ is a $\mathbf{A}$-subnetwork, then $f_{\mathbf{W}}$ is a singular point of $\mathcal{M}_{\mathbf{d},\sigma}$.*

*Proof.* We prove the claim for a generic $\mathbf{A}$-subnetwork $\mathbf{W}$. This is sufficient since the singular locus is Zariski closed. For simplicity of notation, we assume $d_L = 1$—the same proof extends to the general case. Consider the index $i \in \{0, \ldots, L-2\}$ provided by the hypothesis. For $j \in A_{i+1}$ and $k \notin A_{i+2}$, the chain rule implies:

$$\frac{\partial \varphi}{\partial W_{i+2}[k,j]}(\mathbf{W}) = \lambda \, \sigma \left( \sum_{t \notin A_i} W_{i+1}[j,t] \, \sigma \circ f_{\mathbf{W}'}[t] \right), \tag{6}$$

---

[1] In particular, it must hold that $r > \dim(\mathcal{M}_{\mathbf{d},\sigma})$.

where $\lambda$ is a polynomial, and $\mathbf{W}' = (W_1, \ldots, W_i)$. This expression defines a vector in the tangent space of $\mathcal{M}_{\mathbf{d},\sigma}$ at $f_{\mathbf{W}}$. Due to the genericity of $\mathbf{W}$, we have that $\lambda \neq 0$.

Now, $f_{\mathbf{W}}$ and $\lambda$ are unaffected when the $j$-th row of $W_{i+1}$ changes. As this row varies, the hypothesis on the coefficients of $\sigma$ implies that the polynomials $\sigma(\sum_{t \notin A_i} W_{i+1}[j,t] z_t)$ over $\mathbb{R}^{d_i - |A_i|}$ span a linear space of dimension larger than $\dim(\mathcal{M}_{\mathbf{d},\sigma})$. Due to the hypothesis on $d_j - |A_j|$ for $j \leq i$, the rank of the weight matrices $W_j$ is at least $d_i - |A_i|$ for generic $W_j$. Since, moreover, the image of the activation function $\sigma$ is, at least, a half-line in $\mathbb{R}$, we conclude that the image of $\sigma \circ f_{\mathbf{W}'}$ has non-empty interior in $\mathbb{R}^{d_i - |A_i|}$ in the Euclidean topology. By irreducibility of the Zariski topology, this set is Zariski dense. In other words, $\sigma \circ f_{\mathbf{W}'}$ is a dominant map onto $\mathbb{R}^{d_i - |A_i|}$ in the Zariski topology. This implies that if some polynomials over $\mathbb{R}^{d_i - |A_i|}$ are linearly independent, then they remain such over $\mathbb{R}^{d_0}$ after pre-composing them by $\sigma \circ f_{\mathbf{W}'}$—this follows from the general principle of duality in algebraic geometry (Dieudonne & Grothendieck, 1971, I, Corollary 1.2.7). In conclusion, the polynomials from equation 6 generate a linear space of dimension larger than $\dim(\mathcal{M}_{\mathbf{d},\sigma})$, implying that $f_{\mathbf{W}}$ is a singular point. $\qquad\square$

Theorem 4.2 leaves open the question of whether *all* the singularities of the neuromanifold are parametrized by subnetworks—see Section 5 for a discussion. As mentioned above, this is true for linear MLPs, i.e., when $\sigma(x) = x$. Moreover, we provide an explicit example in the appendix (Section D.1), where we manually compute the singular locus (of the Zariski closure of the neuromanifold) for a small network with a cubical activation, and show that subnetworks exhaust all singularities.

*Remark* 4.1. Theorem 4.1 and 4.2 can be potentially extended beyond polynomial activations via polynomial approximation, as anticipated in Section 1. We sketch an informal argument here – a rigorous proof would require details from functional analysis, and we leave it for future investigation (see Section 5). Both theorems are based on showing that some derivatives of $\varphi$ are linearly independent. Namely, Theorem 4.1 relies on showing that the Jacobian $\mathbf{J}_{\mathbf{W}}\varphi$ is full-rank (for generic $\mathbf{W}$), while the proof of Theorem 4.2 argues that some derivatives of $\varphi$ at parameters in the same fiber are linearly independent (see equation 6). Now, suppose that $\sigma$ is not a polynomial, but belongs to a space $\mathcal{F}$ of smooth functions where polynomials (of large degree) are dense. Examples are the space of smooth functions on a compact domain with uniform norm, where density follows from the Stone-Weierstraß approximation theorem, or the space of functions that are analytic around $0$ with the $\ell^2$-distance on their Taylor coefficients, which can be approximated by polynomials by truncating their Taylor series. It is natural to expect that linear independence is an an open condition in $\mathcal{F}$. Indeed, since linear independence of functions is guaranteed when their Wronskian is non-vanishing (Bender & Orszag, 2013), openness follows assuming that the Wronskian is continuous over $\mathcal{F}$. This means that if the theorems hold for an activation $\sigma \in \mathcal{F}$, then they hold in a neighborhood of $\sigma$ (with respect to the topology of $\mathcal{F}$). Since we know that they hold for generic polynomials of large degree, which are dense in $\mathcal{F}$, we conclude that the theorems hold in a dense open subset of $\mathcal{F}$, i.e., for 'almost all' $\sigma \in \mathcal{F}$ (in an appropriate functional sense, depending on the topology of $\mathcal{F}$).

### 4.3 EXPOSEDNESS OF MLPs

Our next main result is concerned with the role of subnetworks in the optimization process. Motivated by Section 3.3, we consider objectives of the form $\mathcal{L}_u(f) = Q(f - u)$, where $f, u \in \mathcal{V}$ and $Q$ is a non-degenerate quadratic form over $\mathcal{V}$. We now introduce a central notion concerned with biases of the optimization process towards subsets of the parameter space $\mathcal{W}$. The notion is general, since it applies to any algebraic map $\varphi \colon \mathcal{W} \to \mathcal{V}$. In particular, it can be used for any polynomial machine learning model, including neural networks with polynomial activations of arbitrary architecture.

**Definition 5.** A subset $S \subseteq \mathcal{W}$ is *critically exposed* if the set

$$U_S = \{u \in \mathcal{V} \mid \exists \mathbf{W} \in S \quad \nabla(\mathcal{L}_u \circ \varphi)(\mathbf{W}) = 0\} \tag{7}$$

has a non-empty interior in $\mathcal{V}$.

In the above, the notion of interior is intended with respect to the Euclidean topology of $\mathcal{V}$, since the latter is the standard one in applications. However, all the following results will hold in the Zariski topology of $\mathcal{V}$. As discussed in Section 3.1, this will result in stronger statements. In fact, by irreducibility of the Zariski topology, if $U_S$ has non-empty interior, then it is dense. In this case,

intuitively, 'almost all' $u$'s belong to it, meaning that $S$ will contain critical points almost certainly. Therefore, from now on we stick to the Zariski topology.

Intuitively, the notion of critically exposedness formalizes the presence of a bias towards $S$ in the optimization process. Since $u$ depends on the dataset, the weights in a critically exposed set are equilibria of the training dynamics for a non-negligible amount of data. More concretely, if $u$ is sampled randomly from a full-support distribution over $\mathcal{V}$, then the weights in $S$ will be equilibria with positive probability.

*Remark* 4.2. If the Jacobian $\mathbf{J_W}\varphi$ of $\varphi$ at $\mathbf{W}$ vanishes at some weights $\mathbf{W} \in \mathcal{W}$, then $\mathbf{W}$ is critical for every $u \in \mathcal{V}$ (actually, for any loss function). In particular, the singleton $S = \{\mathbf{W}\}$ is critically exposed, and $U_S = \mathcal{V}$. In the case of MLPs with $\sigma(0) = 0$, this holds for the zero weights $\mathbf{W} = 0$.

Exposedness can be rephrased in geometric terms. From the discussion at the end of Section 3.3 it follows that $\mathbf{W} \in \mathcal{W}$ is a critical point of $\mathcal{L}_u$ if, and only if, the image of $\mathbf{J_W}\varphi$ is orthogonal to $f_{\mathbf{W}} - u$, according to the scalar product over $\mathcal{V}$ induced by $Q$. In other words, the locus of $u$ for which a given $\mathbf{W}$ is critical coincides with the translated orthogonal complement $f_{\mathbf{W}} + \mathrm{im}(\mathbf{J_W}\varphi)^{\perp}$ of the image of the Jacobian. As a consequence,

$$U_S = \bigcup_{\mathbf{W} \in S} f_{\mathbf{W}} + \mathrm{im}(\mathbf{J_W}\varphi)^{\perp}. \tag{8}$$

The right-hand side is the union of a family of affine subspaces of $\mathcal{V}$ indexed by $S$. Exposedness amounts to the statement that this union has full dimension $\dim(U_S) = \dim(\mathcal{V})$. Below, we leverage on this geometric strategy to show that strict subnetworks of MLPs are critically exposed. As we shall see in the next section, this drastically differs for convolutional architectures, for which the dimensionality of the union $U_S$ is typically deficient.

**Theorem 4.3.** *Suppose $\sigma(0) = 0$. For every $i = 1, \ldots, L-1$, fix $A_i \subseteq \{1, \ldots, d_i\}$, with $|A_i| < d_i$. Let $S \subseteq \mathcal{W}$ be the set of strict $\mathbf{A}$-subnetworks. Then any open set of $S$ is critically exposed.*

*Proof.* Fix a strict $\mathbf{A}$-subnetwork $\mathbf{W}$. From the chain rule applied to equation 1, it follows that

$$\frac{\partial \varphi}{\partial W_k[i,j]}(\mathbf{W}) = \begin{cases} 0 & \text{if } i \in A_k \text{ or } j \in A_{k-1}, \\ \frac{\partial \varphi|_S}{\partial W_k[i,j]}(\mathbf{W}) & \text{otherwise.} \end{cases} \tag{9}$$

In the above, $\varphi|_S \colon S \to \mathcal{V}$ denotes the restriction of $\varphi$ to $S$. Therefore, $\nabla(\mathcal{L}_u \circ \varphi)(\mathbf{W})$ is obtained by padding $\nabla(\mathcal{L}_u \circ \varphi|_S)(\mathbf{W})$ with vanishing entries. In particular, if $\mathbf{W}$ is a critical point for $\mathcal{L}_u \circ \varphi|_S$, it is critical for $\mathcal{L}_u \circ \varphi$ as well. Hence, $\mathrm{im}(\mathbf{J_W}\varphi|_S)^{\perp} = \mathrm{im}(\mathbf{J_W}\varphi)^{\perp}$, and so the union $U_S$ in equation 8 contains the embedded normal bundle of (the smooth locus of) $\varphi(S)$. This bundle has full dimension $\dim(\mathcal{V})$, even when restricted to the image of an open set $T \subseteq S$. This shows that $U_T$ is full-dimensional, and exposedness of $T$ follows. $\qquad\square$

*Remark* 4.3. Theorem 4.3 holds also for activations with $\sigma(0) \neq 0$, if one uses a different definition of strict $\mathbf{A}$-subnetworks $\mathbf{W}$: namely, for every layer $i$ and every $j \in A_i$, the $j$-th column of $W_{i+1}$ vanishes and the $j$-th row of $W_i$ coincides with its $k$-th row for some $k \notin A_i$. These repeated rows in the weight matrices lead to repeated columns in the Jacobian $\mathbf{J_W}\varphi$, instead of 0-columns as in equation 9.

The proof of Theorem 4.3 shows actually more: since $\mathbf{J_W}\varphi$ is obtained by adding vanishing columns to $\mathbf{J_W}\varphi|_S$, the gradient descent dynamics of $\mathcal{L} \circ \varphi$ and of $\mathcal{L} \circ \varphi|_S$ are isomorphic over $S$ for any differentiable loss $\mathcal{L} \colon \mathcal{V} \to \mathbb{R}$. Put simply, from a dynamical perspective, strict subnetworks are equivalent when seen as embedded in $\mathcal{W}$ or as MLPs with a smaller architecture. Similar considerations, with analogous Jacobian arguments, have been made in (Chen et al., 2023).

### 4.4 COMPARISON WITH CONVOLUTIONAL NETWORKS

So far, we have considered neural networks with a fully-connected architecture. In this section, we instead discuss *convolutional* networks—a classical architecture originated in computer vision (Fukushima, 1979; LeCun et al., 1995). We start by recalling the basic definitions. To this end, fix positive integers $k, s, d' \in \mathbb{N}$ representing filter size, stride, and output dimension respectively. The

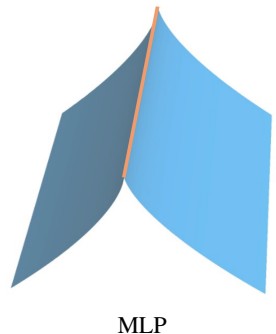
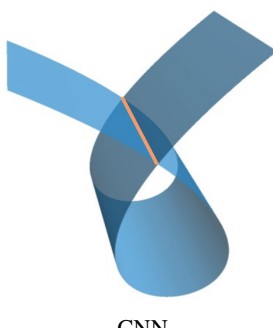

MLP                                                                CNN

Figure 2: Illustration of the different types of singularities (orange) arising in the neuromanifolds of MLPs and CNNs.

convolution between a filter $w \in \mathbb{R}^k$ and an input vector $x \in \mathbb{R}^d$, with $d = s(d' - 1) + k$, is the vector $w \star_s x \in \mathbb{R}^{d'}$ defined for $1 \leq i \leq d'$ as:

$$(w \star_s x)[i] = \sum_{j=1}^{k} w[j] \, x[s(i-1) + j]. \tag{10}$$

Note that we consider one-dimensional convolutions for simplicity of notation, but the theory holds similarly for higher-dimensional ones. Indeed, our arguments build upon Shahverdi et al. (2025), whose results apply verbatim, with the same proofs, to higher-dimensional convolutions.

Now, fix sequences $\mathbf{k}, \mathbf{s} \in \mathbb{Z}_{\geq 1}^L$, $\mathbf{d} \in \mathbb{Z}_{\geq 1}^{L+1}$ such that $d_i = s_i(d_{i+1} + k_i)$ for all $i$, and vectors $\mathbf{w} = (w_1, \ldots, w_L) \in \bigoplus_{i=1}^{L} \mathbb{R}^{k_i}$.

**Definition 6.** A *Convolutional Neural Network* (CNN) with architecture $(\mathbf{k}, \mathbf{s}, \mathbf{d})$ and weights $\mathbf{w}$ is the map $f_{\mathbf{w}} \colon \mathbb{R}^{d_0} \to \mathbb{R}^{d_L}$ given by:

$$f_{\mathbf{w}}(x) = w_L \star_{s_L} \sigma \left( \cdots \star_{s_2} \sigma(w_1 \star_{s_1} x) \right), \tag{11}$$

where $\sigma$ is applied coordinate-wise.

Similarly to MLPs, via abuse of notation, we denote by $\mathcal{W} = \bigoplus_{i=1}^{L} \mathbb{R}^{k_i}$ the parameter space, by $\varphi \colon \mathcal{W} \ni \mathbf{w} \mapsto f_{\mathbf{w}}$ the parametrization map, and its image—i.e., the neuromanifold—by $\mathcal{M}_{\mathbf{k}, \mathbf{s}, \mathbf{d}, \sigma}$.

The geometry of the neuromanifold is well understood for monomial activation functions $\sigma(x) = x^r$ (Shahverdi et al., 2025; Kohn et al., 2022). Below, we extend the main results from Shahverdi et al. (2025) to general polynomial activations with $\sigma(0) = 0$. Specifically, we show that $\varphi$ is regular in $\mathcal{W} \setminus \varphi^{-1}(0)$, meaning that its Jacobian $\mathbf{J}_{\mathbf{w}}\varphi$ has full rank for every $\mathbf{w} \in \mathcal{W} \setminus \varphi^{-1}(0)$. The reason we exclude the fiber of the zero function is that, similarly to MLPs, the Jacobian is rank deficient at $\mathbf{w}$ if $f_{\mathbf{w}} = 0$ (see Remark 4.2).

**Theorem 4.4.** *Let $\sigma$ be a generic polynomial of large enough degree $r \gg 0$ (depending on $L$) with $\sigma(0) = 0$. Then the parametrization map $\varphi$ restricted to $\mathcal{W} \setminus \varphi^{-1}(0)$ is regular, generically one-to-one, and its remaining fibers are finite. In particular, $\dim(\mathcal{M}_{\mathbf{k}, \mathbf{s}, \mathbf{d}, \sigma}) = \dim(\mathcal{W}) = \sum_{i=1}^{L} k_i$.*

The proof is provided in Appendix C.1. Theorem 4.4 establishes a stronger property than the one satisfied by the parametrization of an MLP (cf. Theorem 4.1). As such, it has important consequences in terms of singularities and exposedness. Specifically, since the parametrization $\varphi$ is regular, it does not induce spurious critical points of the loss in parameter space (Trager et al., 2020), i.e., all critical parameters $\mathbf{w}$ (away from $\varphi^{-1}(0)$) yield critical functions $f_{\mathbf{w}}$. Since moreover all fibers of $\varphi$ (excluding 0) are finite, we conclude that the singular points of $\mathcal{M}_{\mathbf{k}, \mathbf{s}, \mathbf{d}, \sigma} \setminus \{0\}$ are *nodal*, i.e., they arise from self-intersections. Differently from the case of MLPs, these singularities are of mild type—see Figure 2 for an illustration, and do not cause critically exposedness. In fact, we now show that, once the fiber of 0 has been excluded, no algebraic set can be critically exposed. This paints a completely different picture than in the case of MLPs (cf. Theorem 4.3).

**Proposition 4.5.** *Assume the hypotheses of Theorem 4.4. Let $S \subset \mathcal{W}$ be an open set of an algebraic variety strictly contained in $\mathcal{W}$. If $0 \notin \varphi(S)$, then $S$ is not critically exposed.*

The proof is provided in Appendix C.3. The fact that, differently from MLPs, subnetworks do not define equilibria for the training dynamics of CNNs has been sometimes observed empirically. For example, it has been established that initializing CNNs with almost all vanishing weights does not result in a collapsed dynamics. Instead, the network recovers from the initialization, and learns well (Blumenfeld et al., 2020).

Nevertheless, subnetworks of CNNs can yield singular points of the neuromanifold. In fact, we show now that *all* singularities arise from subnetworks in the case of convolutional architectures. Since CNNs exhibit a weight-sharing pattern, the notion of subnetwork is different than for MLPs. Specifically, we wish subnetworks to be reparametrizable by a smaller architecture, i.e., by smaller filters. We will think of the latter as being represented by filters that are padded by vanishing entries on the left or on the right. This leads to the following definition. Pick integers $A_1, \ldots, A_L$ such that $0 \leq A_i \leq k_i$ for every $i$, and define $\mathbf{A} = (A_1, \ldots, A_L)$.

**Definition 7.** A parameter $\mathbf{w} \in \mathcal{W}$ is an $\mathbf{A}$-subnetwork if for each $i = 1, \ldots, L$, $w_i[j] = 0$ holds either for all $j = 1, \ldots, A_i$ or for all $j = k_i - A_i + 1, \ldots, k_i$. The subnetwork is proper if $A_i > 0$ for at least one $i$.

Given an $\mathbf{A}$-subnetwork, we denote by $t_i \in \mathbb{Z}$ the cardinality of $A_i$, equipped with a positive or negative sign corresponding to whether $w_i$ is padded with zeros on the left or on the right, respectively. We also recursively define $\tilde{t}_0 = 0$, $\tilde{t}_i = t_i + \tilde{t}_{i-1}/s_{i-1}$ for $i \geq 1$. The proof of the following claim is provided in Appendix C.2, alongside an example illustrating the condition on the $\tilde{t}_i$ in Appendix D.2.

**Theorem 4.6.** *Assume the hypotheses of Theorem 4.4. Let $\mathbf{w} \in \mathcal{W} \setminus \varphi^{-1}(0)$. Then $f_{\mathbf{w}}$ is a singular point of $\mathcal{M}_{\mathbf{k}, \mathbf{s}, \mathbf{d}, \sigma}$ if, and only if, $\mathbf{w}$ is a proper $\mathbf{A}$-subnetwork such that $\tilde{t}_i \in \mathbb{Z}$ for all $i$, and $\tilde{t}_L = 0$.*

## 5 CONCLUSIONS, LIMITATIONS, AND FUTURE WORK

We have considered neural networks with an activation function that is a sufficiently-generic polynomial. By leveraging arguments from algebraic geometry, we have studied questions related to identifiability, singularity, and exposedness for MLPs and CNNs. Overall, this work addresses open problems in neuroalgebraic geometry (Marchetti et al., 2024), and provides a novel geometric perspective on the sparsity bias of neural networks. Yet, it is subject to limitations, as outlined below.

**Complete characterization of singularities and exposedness.** Theorems 4.2 states that subnetworks of MLPs can parametrize singular points of the neuromanifold. However, it is unclear whether this describes all singularities. For deep linear MLPs (Trager et al., 2020) and for deep polynomial CNNs (see Theorem 4.6), all singularities come from subnetworks. The problem of a full characterization of the singularities of polynomial MLPs is left open. Similarly, a complete classification of critically exposed parameter subsets for polynomial MLPs, beyond the subnetworks described in Theorem 4.3 and Remark 4.3, is crucial for a complete understanding of the biases of deep networks.

**Type of critical points.** Our analysis has only considered whether weights are critical points of the objective, disregarding the type of criticality (local minimum/maximum, or saddle). Although all critical points are equilibria of the dynamics, local minima correspond to the (local) attractors. Thus, it would be interesting to incorporate the type of critical points in our analysis; in particular, to understand whether subnetworks are local minima for non-negligible amounts of $u \in \mathcal{V}$.

**Beyond the algebraic.** Throughout this work, in order to leverage on tools from algebraic geometry, we have focused on neural networks with polynomial activations. However, several popular activation functions are either piece-wise polynomial (e.g., ReLU), or completely non-algebraic (e.g., Tanh and SoftMax). Extending our results to the non-polynomial case is therefore important. As mentioned in Section 1, a promising strategy to this end is polynomial approximation; since polynomials can (locally) approximate arbitrary continuous functions, neuromanifolds of general neural networks can be approximated by algebraic ones. This approach is, generally speaking, part of the research program of neuroalgebraic geometry (Marchetti et al., 2024), and has been often fruitful for extending results for polynomial networks beyond the algebraic domain (Boullé et al., 2020; Zhang & Kileel, 2023). To this end, we have sketched an argument in Remark 4.1. However, a rigorous proof comprising all the functional analysis details is left for future investigation.

ACKNOWLEDGEMENTS

This work was partially supported by the Wallenberg AI, Autonomous Systems and Software Program (WASP) funded by the Knut and Alice Wallenberg Foundation.

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

APPENDIX

## A SINGULARITY AND CRITICALITY

In this section, we aim to clarify the distinction between singular points of a variety and critical points of its parametrization. Although both notions involve degeneracies, they can arise from different mechanisms and have different implications for optimization. Singular points are geometric features of the variety itself, reflected in the dimension of the tangent space, while critical points depend on the differential of the chosen parametrization.

To formally define singularities of a variety $X \subseteq \mathbb{R}^n$, we assume that $X$ is *irreducible* (i.e., it cannot be written as the union of two proper non-empty subvarieties) and consider its *vanishing ideal* $I_X$ that consists of all polynomials in $\mathbb{R}[x_1, \ldots, x_n]$ that vanish along $X$. We then fix a generating set $\langle f_1, \ldots, f_s \rangle = I_X$ for that ideal and compute the $s \times n$ Jacobian matrix $\mathbf{J}$ whose $(i, j)$-th entry is $\frac{\partial f_i}{\partial x_j}$. For almost every point $p$ on $X$ (i.e., except for $p$ on some proper subvariety), the rank of $\mathbf{J}(p)$ is the same. Those are the *smooth* points of $X$, and at those points $p$, the kernel of $\mathbf{J}(p)$ is the tangent space of $X$ at $p$. At the remaining points on $X$, the rank of the Jacobian drops; those are the *singular* points. They form a subvariety of $X$, defined by the ideal $I_X$ and the $c \times c$ minors of the Jacobian, where $c$ is the rank of $\mathbf{J}(p)$ at a smooth point $p$ of $X$.

Through the classical examples of a nodal and a cuspidal cubic curve, we now illustrate how the concepts of singularities on varieties and critical points of parametrizations differ, and how tangent spaces play a key role in their characterization.

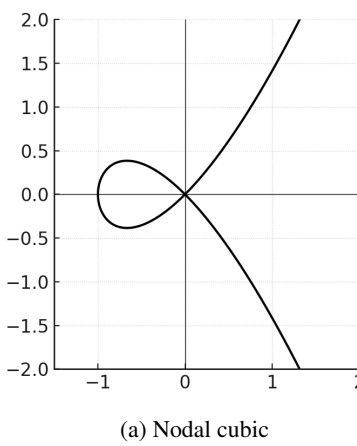
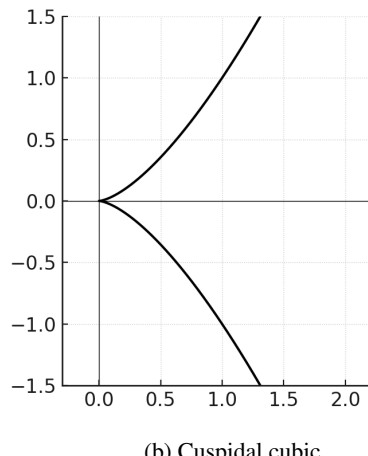

(a) Nodal cubic

(b) Cuspidal cubic

Figure 3: Two singular cubic curves.

**Example A.1.** Consider the parametrization of two singular cubic curves:

$$\varphi \colon \mathbb{R} \to \mathbb{R}^2, \qquad t \mapsto (t^2 - 1, \ t(t^2 - 1)) \qquad \text{(nodal cubic } y^2 = x^2(x + 1)), \qquad (12)$$

$$\psi \colon \mathbb{R} \to \mathbb{R}^2, \qquad t \mapsto (t^2, \ t^3) \qquad \text{(cuspidal cubic } y^2 = x^3). \qquad (13)$$

In both cases, the origin is a singular point of the curve; see Figure 3. For the nodal cubic, every point on the curve except the origin has a fiber of size one. At the origin, however, the fiber is unusual, since it contains two distinct parameters: $\varphi^{-1}(0, 0) = \{-1, 1\}$. In contrast, the cuspidal parametrization is injective, i.e., $\psi^{-1}(x, y)$ consists of a single point for every $(x, y) \in \text{im}(\psi)$.

The Jacobians of both parametrizations are

$$\mathbf{J}_t \varphi = [2t \quad 3t^2 - 1], \qquad (14)$$

$$\mathbf{J}_t \psi = [2t \quad 3t^2]. \qquad (15)$$

Thus, $\varphi$ is regular at every $t$ (i.e., the Jacobian never vanishes), while $\psi$ is critical at $t = 0$ (where the Jacobian is $[0, 0]$). In other words, the cuspidal parametrization is not regular at the cusp, although it does not have any unusual fibers; its singularity arises solely from the drop in rank of the Jacobian.

To see algebraically that the origin is indeed the only singular point of both curves, we compute the Jacobian of their defining equations $f(x, y) = x^2(x + 1) - y^2$ and $g(x, y) = x^3 - y^2$, respectively:

$$\mathbf{J}f = [3x^2 + 2x \quad - 2y], \tag{16}$$

$$\mathbf{J}g = [3x^2 \quad - 2y]. \tag{17}$$

Both vanish precisely when $(x, y) = (0, 0)$. At all other points on the curves, the kernel of the Jabobian is the 1-dimensional tangent space. At the singularity $(0, 0)$, the Zariski tangent space is the whole 2-dimensional ambient plane $\mathbb{R}^2$. For the node, the plane is spanned by the two distinct tangent directions corresponding to $t = -1$ and $t = 1$, with slopes 1 and $-1$, respectively. For the cusp, however, the 2-dimensional tangent space arises from the vanishing of all first-order terms, which corresponds geometrically to infinite curvature at the origin. In particular, since $\mathbf{J}_0\psi = 0$, this tangent space cannot be detected solely from the diffential of the parametrization. $\diamond$

This example illustrates that critical points of a parametrization can cause singularities on the image variety, but that not all singularities of parametrized varieties arise like that. Moreover, not every critical point of a parametrization leads to a singular point on the image. This can be for instance seen by projecting the cuspidal curve in Figure 3b onto $y$-axis: Composing the cuspidal parametrization $\psi$ with that projection leads to a parametrization $t \mapsto t^3$ of the line $\mathbb{R}^1$ (which is clearly smooth everywhere) that is not regular at $t = 0$. All in all, critical points of parametrizations and singularities of their image varieties are subtly related, but non implies the other in general.

The effect that different types of singularities have when optimizing a loss over high-dimensional algebraic varieties is largely unknown. Cuspidal-type singularities, arising from a rank drop of the parametrization's Jacobian, may appear both as spurious (Trager et al., 2020) and genuine critical points of the loss. In the case of MLPs, we show that these types of singularities are critically exposed. In contrast, nodal-type singularities do not exhibit this behavior. This reflects the main difference between the singular points of MLPs and CNNs (see Figure 2) since we show that all singularities of CNNs are of nodal type.

## B  PROOF OF THEOREM 4.1

In this section, we prove Theorem 4.1. We begin with some technical results.

**Lemma B.1.** *Let $\sigma(x) = x^{\beta_L} + \cdots + x^{\beta_1}$ be a polynomial activation function such that $\beta_j > \beta_{j-1}^{L-1}$ for every $j > 1$, and $\beta_1 > 1$. Define $\sigma_{\beta_i}(x) = x^{\beta_i}$. Then for all weights $\mathbf{W} \in \mathcal{W}$, the MLP can be decomposed as:*

$$f_{\mathbf{W}}(x) = \sum_{i=1}^{L} W_L \sigma_{\beta_i} W_{L-1} \sigma_{\beta_i} W_{L-2} \cdots \sigma_{\beta_i} W_1(x) + R(x), \tag{18}$$

*where the remainder $R(x)$ does not contain any monomial of degree $\beta_j^{L-1}$ for every $j = 1, \ldots, L$.*

*Proof.* It suffices to show that monomials of degree $\beta_j^{L-1}$ appear solely in the term

$$W_L \sigma_{\beta_j} W_{L-1} \sigma_{\beta_j} W_{L-2} \cdots \sigma_{\beta_j} W_1(x). \tag{19}$$

We verify this by tracking the degrees generated layer by layer.

First, note that the monomials of degrees $\beta_1^{L-1}$ and $\beta_L^{L-1}$ are clearly only generated by the terms

$$W_L \sigma_{\beta_1} W_{L-1} \sigma_{\beta_1} W_{L-2} \cdots \sigma_{\beta_1} W_1(x) \quad \text{and} \quad W_L \sigma_{\beta_L} W_{L-1} \sigma_{\beta_L} W_{L-2} \cdots \sigma_{\beta_L} W_1(x), \tag{20}$$

respectively. Thus, we focus our analysis on degrees $\beta_j^{L-1}$ for $1 < j < L$.

We proceed by induction on the number of layers $k$, where $2 \leq k \leq L$. Specifically, we claim that at layer $k$, monomials of degree $\beta_j^{k-1}$ appear exclusively through the composition

$$W_k \sigma_{\beta_j} W_{k-1} \sigma_{\beta_j} \cdots \sigma_{\beta_j} W_1(x). \tag{21}$$

Moreover, we claim that the closest degrees immediately preceding and succeeding $\beta_j^{k-1}$ are given by $(\beta_j^{k-2} - 1)\beta_j + \beta_{j-1}$ and $(\beta_1^{k-2} - 1)\beta_1 + \beta_{j+1}$, respectively.

For $k = 2$, note that the degrees appearing at the second layer are

$$\beta_1 < \beta_2 < \cdots < \beta_{L-1} < \beta_L, \tag{22}$$

and thus $R(x) = 0$ at this stage.

For the induction hypothesis, assume that the claim holds for some layer $k < L$. That is, monomials of degrees $\beta_j^{k-1}$ are solely generated by equation 21 at layer $k$, and the sequence of degrees generated has the form

$$\beta_1^{k-1} < \beta_1^{k-1} - \beta_1 + \beta_2 < \cdots < \beta_i^{k-1} - \beta_i + \beta_{i-1} < \beta_i^{k-1}$$
$$< \beta_1^{k-1} - \beta_1 + \beta_{i+1} < \cdots < \beta_L^{k-1} - \beta_L + \beta_{L-1} < \beta_L^{k-1}, \tag{23}$$

where intermediate terms represent monomials whose degrees lie strictly between the bounds above.

For the inductive step, consider the $(k + 1)$-layer composition, expressed by

$$W_{k+1}\sigma W_k \sigma \cdots \sigma W_1(x) = \sum_{i=1}^{L} W_{k+1}\sigma_{\beta_i} W_k \sigma \cdots \sigma W_1(x). \tag{24}$$

To obtain monomials of degree exactly $\beta_i^k$, we analyze each summand individually. For any choice of $\sigma_{\beta_j}$ with $j \neq i$, we claim that no monomial of degree $\beta_i^k$ emerges. Hence, the only valid choice is $j = i$. To see this, we consider two cases

- $j < i$ : In this case, to obtain a term of degree at least $\beta_i^k$ in the multinomial expansion of $\sigma_{\beta_j} W_k \sigma \ldots \sigma W_1(x)$, we have to use at least one the monomials of $W_k \sigma \ldots \sigma W_1(x)$ of degree larger than $\beta_i^{k-1}$. By equation 23, the smallest possible degree is $\beta_i^{k-1} - \beta_1 + \beta_{i+1}$. However, by our assumption, this is strictly larger than $\beta_i^{L-1} \geq \beta_i^k$.

- $j > i$ : In this case, since the smallest degree appearing in $W_k \sigma \ldots \sigma W_1(x)$ is $\beta_1^{k-1}$ according to equation 23, the smallest degree after passing through $\sigma_{\beta_j}$ is $\beta_j \beta_1^{k-1} > \beta_j > \beta_i^{L-1}$. Hence, all terms are of degree larger than $\beta_i^k$.

Moreover, the neighboring degrees of $\beta_i^k$ in $\sigma_{\beta_i} W_k \sigma \cdots \sigma W_1(x)$ are

$$\underbrace{\beta_i^{k-1}(\beta_i - 1) + (\beta_i^{k-1} - \beta_i + \beta_{i-1})}_{=\beta_i^k - \beta_i + \beta_{i-1}} < \beta_i^k < \underbrace{\beta_1^{k-1}(\beta_1 - 1) + (\beta_1^{k-1} - \beta_1 + \beta_{i+1})}_{=\beta_1^k - \beta_1 + \beta_{i+1}}. \tag{25}$$

This exactly confirms the inductive step, hence the uniqueness of the decomposition in equation 18. $\square$

**Lemma B.2.** *Let $e_1 > e_2$ be positive integers of opposite parity. For $i = 1, \ldots, L$, let $W_i \in \mathbb{R}^{d_i \times d_{i-1}}$ be generic. For $j = 1, 2$, define*

$$A_j := \left\{ \left(D_{j,1}W_1, \, D_{j,2}W_2 D_{j,1}^{-e_j}, \, \ldots, \, W_L D_{j,L-1}^{-e_j}\right) \mid D_{j,k} \in \mathrm{diag}_{d_k}^{\times}(\mathbb{R}) \; \forall k = 1, \ldots, L-1 \right\}, \tag{26}$$

*where $\mathrm{diag}_{d_k}^{\times}(\mathbb{R})$ is the set of real invertible diagonal matrices of size $d_k \times d_k$. Then $A_1 \cap A_2$ consists exactly of those tuples in $A_1$ for which each $D_{1,k} = \lambda_{1,k}I$ is a multiple of the identity for some $\lambda_{1,k} \in \mathbb{R} \setminus \{0\}$ that satisfy*

$$\lambda_{1,1}^{e_2^{L-2}} \lambda_{1,2}^{e_2^{L-3}} \cdots \lambda_{1,L-2}^{e_2} \lambda_{1,L-1} = 1 \tag{27}$$

*Proof.* Take $(\widehat{W}_1, \ldots, \widehat{W}_L) \in A_1 \cap A_2$. By definition, there exist diagonal invertible matrices $D_{j,k}$ such that for $k = 2, \ldots, L - 1$:

$$\widehat{W}_1 = D_{1,1}W_1 = D_{2,1}W_1, \qquad \widehat{W}_k = D_{1,k}W_k D_{1,k-1}^{-e_1} = D_{2,k}W_k D_{2,k-1}^{-e_2}. \tag{28}$$

Since a generic matrix admits no diagonal symmetries, $D_{1,1} = D_{2,1} =: D$. For $k = 2$ the second equality becomes

$$D_{2,2}^{-1} D_{1,2} W_2 = W_2 D^\Delta, \tag{29}$$

where $\Delta := e_1 - e_2$. Left multiplication by a diagonal matrix scales rows, while right multiplication scales columns. Thus, since $W_2$ is generic, both $D^\Delta$ and $D_{2,2}^{-1} D_{1,2}$ are the same scalar multiple of the identity. Since $\Delta$ is odd, this yields $D = \lambda_{1,1} I$ and $D_{2,2} = \lambda_{1,1}^{-\Delta} D_{1,2}$ for some $\lambda_{1,1} \in \mathbb{R} \setminus \{0\}$.

By plugging this relation into the equality for $k = 3$ and repeating the same argument shows that $D_{1,2} = \lambda_{1,2} I$, and that $D_{2,3} = \lambda_{1,1}^{-e_2 \Delta} \lambda_{1,2}^{-\Delta} D_{1,3}$. Inductively, for every $k = 1, \ldots, L-1$ we get

$$D_{1,k-1} = \lambda_{1,k-1} I, \qquad D_{2,k} = \lambda_{1,1}^{-\Delta e_2^{k-2}} \lambda_{1,2}^{-\Delta e_2^{k-3}} \cdots \lambda_{1,k-1}^{-\Delta} D_{1,k}. \tag{30}$$

At the last layer, the equality $W_L D_{1,L-1}^{-e_1} = W_L D_{2,L-1}^{-e_2}$ gives $D_{2,L-1} = D_{1,L-1}^{e_1/e_2}$. Using the equality for $k = L-1$ in equation 30, we get

$$D_{1,L-1}^{\Delta/e_2} = \lambda_{1,1}^{-\Delta e_2^{L-3}} \lambda_{1,2}^{-\Delta e_2^{L-4}} \cdots \lambda_{1,L-2}^{-\Delta} I. \tag{31}$$

Hence, $D_{1,L-1}$ is also a scalar multiple of the identity matrix, say $\lambda_{1,L-1} I$, and thus

$$\lambda_{1,1}^{e_2^{L-2}} \lambda_{1,2}^{e_2^{L-3}} \cdots \lambda_{1,L-2}^{e_2} \lambda_{1,L-1} = 1. \tag{32}$$

Conversely, a direct computation reveals that these scalar multiples of the identity yield indeed a point in the intersection $A_1 \cap A_2$. □

Next, we recall a result from literature on identifiability of MLPs with monomial activation functions.

**Theorem B.3** (Finkel et al. (2024)). *Suppose that $\sigma(x) = x^r$, with $r \geq 6m^2 - 6m$, where $m = 2\max\{d_1, \ldots, d_{L-1}\}$. For generic $\mathbf{W} \in \mathcal{W}$, the fiber $\varphi^{-1}(f_{\mathbf{W}})$ consists of weights of the form*

$$\left( P_{i,1} D_{i,1} W_1, \; P_{i,2} D_{i,2} W_2 D_{i,1}^{-\beta_i} P_{i,1}^\top, \; \ldots, \; W_L D_{i,L-1}^{-\beta_i} P_{i,L-1}^\top \right), \tag{33}$$

*where $P_{i,j}$ are permutation matrices and $D_{i,j}$ are invertible diagonal matrices of size $d_j \times d_j$.*

We can finally prove the desired theorem.

*Proof of Theorem 4.1.* We fix a sufficiently large degree $r$. It is sufficient to show that there exists a polynomial $\sigma$ of degree $r$ for which the theorem holds. Indeed, $\varphi$ having finite fibers is equivalent to its Jacobian attaining the maximal rank $\dim(\mathcal{W})$ at generic weights $\mathbf{W}$. The condition that the Jacobian has maximum rank $\dim(\mathcal{W})$ is open with respect to the Zariski topology in the coefficient space of $\sigma$. Since open sets are dense, if this condition holds for one polynomial activation of degree $r$, it holds for a generic one.

Consider the sparse polynomial

$$\sigma(x) := \sum_{i=1}^{L} x^{\beta_i}, \tag{34}$$

where $\beta_i > \beta_{i-1}^{L-1}$ for all $i$ (note that $\beta_L = r$). Set $\sigma_{\beta_i}(x) := x^{\beta_i}$. Then, by Lemma B.1, the output of the MLP uniquely decomposes as

$$\sum_{i=1}^{L} W_L \sigma_{\beta_i} W_{L-1} \sigma_{\beta_i} W_{L-2} \cdots \sigma_{\beta_i} W_1(x) + (\text{remaining terms}). \tag{35}$$

Each monomial term in equation 35 is an MLP with monomial activation $x^{\beta_i}$. By Theorem B.3, if we choose $\beta_1 \geq 6m^2 - 6m$, where $m = 2\max\{d_1, \ldots, d_{L-1}\}$, then the generic fiber of the parametrization map of such an MLP consists of weights of the form

$$\left( P_{i,1} D_{i,1} W_1, \; P_{i,2} D_{i,2} W_2 D_{i,1}^{-\beta_i} P_{i,1}^\top, \; \ldots, \; W_L D_{i,L-1}^{-\beta_i} P_{i,L-1}^\top \right), \tag{36}$$

where $P_{i,j}$ are permutation matrices and $D_{i,j}$ are invertible diagonal matrices of size $d_j \times d_j$. For generic parameters $\mathbf{W}$, the fiber $\varphi^{-1}(f_{\mathbf{W}})$ is contained in the intersection of the sets of such tuples. In other words, every tuple in the fiber $\varphi^{-1}(f_{\mathbf{W}})$ can be expressed as in equation 36 for *every* $i = 1, \ldots, L$.

Since $\mathbf{W}$ is generic and each $D_{i,j}$ is diagonal, it follows that all the corresponding permutation matrices must coincide at the intersection. That is, we have $P_{i,j} = P_{k,j}$ for all $i, j, k$. Since we wish to show finiteness of fibers, without loss of generality, we may assume that these permutation matrices are identities.

Now, by comparing tuples equation 36 for different indices $i$, Lemma B.2 implies that each diagonal matrix reduces to a scalar multiple of the identity, that is, $D_{i,j} = \lambda_{i,j} I$ for some $\lambda_{i,j} \in \mathbb{R}^*$. Moreover, the intersection is characterized explicitly by the polynomial system:

$$
\begin{cases}
\lambda_{L,1}^{\beta_{L-1}^{L-2}} \lambda_{L,2}^{\beta_{L-1}^{L-3}} \cdots \lambda_{L,L-2}^{\beta_{L-1}} \lambda_{L,L-1} - 1 = 0, \\
\quad \vdots \\
\lambda_{L,1}^{\beta_1^{L-2}} \lambda_{L,2}^{\beta_1^{L-3}} \cdots \lambda_{L,L-2}^{\beta_1} \lambda_{L,L-1} - 1 = 0.
\end{cases}
\tag{37}
$$

We use the lattice-ideal approach in toric geometry to show that this system has finitely many solutions; see (Şahin, 2023, Section 7). Let $A[i,j] = \beta_{L-i}^{L-1-j}$. Then $A$ is a Vandermonde matrix of size $(L-1) \times (L-1)$ with determinant

$$
\det(A) = \pm \prod_{1 \le i < j \le L-1} (\beta_j - \beta_i) \neq 0.
\tag{38}
$$

Using the Smith normal form, there exist matrices $U, V \in \mathrm{GL}_{L-1}(\mathbb{Z})$ such that

$$
UAV = \mathrm{diag}(\ell_1, \ldots, \ell_{L-1}),
\tag{39}
$$

where $\ell_1 \mid \ell_2 \mid \cdots \mid \ell_{L-1}$. Letting $\Lambda = (\lambda_{L,1}, \ldots, \lambda_{L,L-1}) \in (\mathbb{C}^*)^{L-1}$, define new coordinates $\Omega = (\omega_1, \ldots, \omega_{L-1})$ via

$$
\omega_j := \prod_{k=1}^{L-1} \lambda_{L,k}^{V^{-1}[j,k]}.
\tag{40}
$$

Since $V$ is unimodular, the map $\Lambda \mapsto \Omega$ is an algebraic automorphism of the torus. In these new coordinates, the system equation 37 is equivalent to $\omega_1^{\ell_1} = 1, \ldots, \omega_{L-1}^{\ell_{L-1}} = 1$, which has exactly $\ell_1 \cdots \ell_{L-1} < \infty$ solutions in $(\mathbb{C}^*)^{L-1}$. Hence, $\varphi^{-1}(f_{\mathbf{W}})$ is finite, and the claimed dimension follows directly from the fiber-dimension theorem. $\square$

## C    POLYNOMIAL CONVOLUTIONAL NETWORKS

In this section, we provide technical details for the proofs in Section 4.4. We first establish a general result on univariate polynomials $\sigma(x) = \sum_{i=0}^{r} a_i x^i$.

**Lemma C.1.** *Suppose that $r > 2$ and that $a_r, a_{r-1} \neq 0$. Moreover, let $u_1, \ldots, u_L \in \mathbb{R} \setminus \{0\}$ and $\lambda_1, \ldots, \lambda_L \in \mathbb{R}$ be such that, for all $x \in \mathbb{R}$,*

$$
\lambda_L u_L \, \sigma\Big(\lambda_{L-1} u_{L-1} \, \sigma\big(\ldots \sigma(\lambda_1 u_1 x) \ldots\big)\Big) = u_L \, \sigma\Big(u_{L-1} \sigma\big(\ldots \sigma(u_1 x) \ldots\big)\Big).
\tag{41}
$$

*Then, $\lambda_1 = \cdots = \lambda_L = 1$.*

*Proof.* Consider the following rational function:

$$
P(x) := \frac{x \, \sigma'(x)}{\sigma(x)},
\tag{42}
$$

which is well-defined wherever $\sigma(x) \neq 0$. Define inductively:

$$
\begin{cases}
G_0(x) := \lambda_1 u_1 x, \\
G_k(x) := \lambda_{k+1} u_{k+1} \sigma(G_{k-1}(x)).
\end{cases}
\qquad
\begin{cases}
H_0(x) := u_1 x, \\
H_k(x) := u_{k+1} \sigma\big(H_{k-1}(x)\big).
\end{cases}
\tag{43}
$$

The left-hand side of equation 41 coincides with $G_{L-1}(x) = \lambda_L u_L \sigma(G_{L-2}(x))$, while the right-hand side is $H_{L-1}(x) = u_L \sigma(H_{L-2}(x))$. Now, by differentiating equation 41 and then dividing by equation 41, we obtain

$$P(G_{L-2}) \frac{G'_{L-2}}{G_{L-2}} = P(H_{L-2}) \frac{H'_{L-2}}{H_{L-2}}. \tag{44}$$

By applying the chain rule iteratively, the above identity is equivalent to

$$\prod_{k=0}^{L-2} P(G_k(x)) = \prod_{k=0}^{L-2} P(H_k(x)). \tag{45}$$

We assume for contradiction that $\lambda_k \neq 1$ for some $k$. Let $m := \min\{\, 0 \leq k \leq L - 2 \mid \lambda_{k+1} \neq 1 \,\}$. For every $j < m$, we have that $G_j = H_j$. Moreover, $G_m(x) = \lambda_{m+1} H_m(x)$ with $\lambda_{m+1} \neq 1$. In particular, equation 45 becomes

$$\prod_{k=m}^{L-2} P(G_k(x)) = \prod_{k=m}^{L-2} P(H_k(x)). \tag{46}$$

Now, set $\beta := -\frac{a_{r-1}}{a_r}$. As $x \to \infty$, the rational function $P$ satisfies $P(x) = r + \frac{\beta}{x} + O(x^{-2})$. Thus, for every $k \geq 0$, we have

$$P(G_k(x)) = r + \frac{\beta}{G_k(x)} + O(x^{-2 \deg G_k}), \qquad P(H_k(x)) = r + \frac{\beta}{H_k(x)} + O(x^{-2 \deg H_k}). \tag{47}$$

Since $\deg G_k = \deg H_k = r^k$, the terms of order up to $-r^m$ in equation 46 are

$$\left( \prod_{k=m+1}^{L-2} r \right) \cdot (r + \frac{\beta}{G_m(x)}) + O(x^{-r^m - 1}) = \left( \prod_{k=m+1}^{L-2} r \right) \cdot (r + \frac{\beta}{H_m(x)}) + O(x^{-r^m - 1}). \tag{48}$$

Using $G_m(x) = \lambda_{m+1} H_m(x)$, we obtain

$$\beta \frac{1 - \lambda_{m+1}}{\lambda_{m+1} H_m(x)} = O(x^{-r^m - 1}). \tag{49}$$

Since the left hand-side is of order $O(x^{-r^m})$ and $\beta \neq 0$ by hypothesis, we conclude that $\lambda_{m+1} = 1$, which is a contradiction $\qquad \square$

**Corollary C.2.** *Suppose that $r > 2$, $a_0 = 0$ and $a_r, a_{r-1} \neq 0$. Moreover, let $w_1 \in \mathbb{R}^{k_1}, \dots, w_L \in \mathbb{R}^{k_L}$ be non-zero filters and let $\lambda_1, \dots, \lambda_L \in \mathbb{R}$ be such that, for all $x \in \mathbb{R}^{d_0}$,*

$$\lambda_L w_L \star_{s_L} \sigma (\cdots \star_{s_2} \sigma(\lambda_1 w_1 \star_{s_1} x)) = w_L \star_{s_L} \sigma (\cdots \star_{s_2} \sigma(w_1 \star_{s_1} x)). \tag{50}$$

*Then, $\lambda_1 = \cdots = \lambda_L = 1$.*

*Proof.* Let $i = 1, \dots, d_0$ be the smallest index such that the variable $x_i$ appears in one of the monomials of equation 50 with non-zero coefficient. Substituting $x_j \mapsto 0$ for all $j \neq i$ in equation 50 yields the equality of univariate polynomials in equation 41, where $u_k$ is the first non-zero entry of the filter $w_k$. Thus, the claim follows from Lemma C.1. $\qquad \square$

## C.1 PROOF OF THEOREM 4.4

*Proof.* We first show regularity. Similarly to the proof of Theorem 4.1, we will do so for a specific activation function, which will imply the statement for a generic one. Consider the polynomial activation defined by

$$\sigma(x) = x^{\beta_L} + \cdots + x^{\beta_1}, \tag{51}$$

where $\beta_1 > 1$, $\beta_L = r$, and $\beta_j > \beta_{j-1}^{(L-1)}$ for all $j > 1$. We will show that the parametrization map $\varphi$ with this activation is regular on $\mathcal{W} \setminus \varphi^{-1}(0)$. Note that $\varphi^{-1}(0)$ consists precisely of those filter tuples where one of the filter is zero, because convolving with a non-zero filter is a full-rank linear map (Shahverdi et al., 2025, Lemma 4.1).

By Lemma B.1, the CNN output decomposes uniquely as

$$f_{\mathbf{w}} = \sum_{i=1}^{L} w_L \star_{s_L} \sigma_{\beta_i} \left( \cdots \star_{s_2} \sigma_{\beta_i} (w_1 \star_{s_1} x) \right) + \text{(remaining terms)}. \tag{52}$$

From (Shahverdi et al., 2025, Proposition A.2), if $f_{\mathbf{w}} \neq 0$, the kernel of the differential of the CNN parametrization with monomial activation $\sigma_{\beta_i}$ is explicitly given by the subspace

$$\mathcal{A}_i := \{ \lambda_{i,1} w_1, (-\lambda_{i,1} \beta_i + \lambda_{i,2}) w_2, \ldots, (-\lambda_{i,L-2} \beta_i + \lambda_{i,L-1}) w_{L-1}, -\lambda_{i,L-1} \beta_i w_L \mid \lambda_{i,j} \in \mathbb{R} \} \tag{53}$$

Using the decomposition equation 52, we deduce that the kernel of the Jacobian $\mathbf{J_w}\varphi$ is contained in the intersection $\bigcap_{i=1}^{L} \mathcal{A}_i$. Now, let $v_j \in \mathbb{R}^L$ contain the coefficients of the filters $w_k$ in $\mathcal{A}_j$ and define

$$\gamma_j := (\beta_j^{L-1}, \beta_j^{L-2}, \ldots, \beta_j, 1)^\top. \tag{54}$$

Then, $\langle \gamma_j, v_j \rangle = 0$. Hence, in $\bigcap_{i=1}^{L} \mathcal{A}_i$, we must have

$$\begin{pmatrix} \beta_L^{L-1} & \beta_L^{L-2} & \cdots & \beta_L & 1 \\ \beta_{L-1}^{L-1} & \beta_{L-1}^{L-2} & \cdots & \beta_{L-1} & 1 \\ \vdots & \vdots & \ddots & \vdots & \vdots \\ \beta_1^{L-1} & \beta_1^{L-2} & \cdots & \beta_1 & 1 \end{pmatrix} v_L = 0. \tag{55}$$

Since $\beta_i - \beta_j \neq 0$ for all $i \neq j$, the matrix in equation 55 is a full-rank Vandermonde matrix, which is invertible. Therefore, the only solution is $v_L = 0$. It follows that $\bigcap_{i=1}^{L} \mathcal{A}_i = \{0\}$, and hence the parameterization map $\varphi$ has an injective differential over $\mathcal{W} \setminus \varphi^{-1}(0)$. In other words, $\varphi$ is an immersion away from $\varphi^{-1}(0)$.

Since $\varphi$ is an immersion at generic weights $\mathbf{w} \in \mathcal{W}$, it necessarily has finite fibers. We wish to show that the generic fiber is a singleton. To this end, choose weights $\mathbf{w} = (w_1, \ldots, w_L)$, where $w_i = (1, 1, \ldots, 1)$ for all $i$. From (Shahverdi et al., 2025, Theorem 4.6), it follows that the fiber $\varphi^{-1}(f_{\mathbf{w}})$ is contained in $\{(\lambda_1 w_1, \ldots, \lambda_L w_L) \mid \lambda_i \in \mathbb{R}\}$. But then Corollary C.2 implies that $\varphi^{-1}(f_{\mathbf{w}}) = \{\mathbf{w}\}$ is a singleton. Since $\varphi$ is regular at $\mathbf{w}$, there is an open neighborhood around $\mathbf{w}$ such that every fiber remains a singleton. Since open sets are dense in the Zariski topology, we conclude that the generic fiber is a singleton. $\square$

## C.2 PROOF OF THEOREM 4.6

*Proof.* Since $\varphi$ is regular, generically one-to-one and has finite fibers over $\mathcal{W} \setminus \varphi^{-1}(0)$, the property that $f_{\mathbf{w}}$ is a singular point is equivalent to the corresponding fiber $\varphi^{-1}(f_{\mathbf{w}})$ not being a singleton.

We start by assuming that $f_{\mathbf{w}} \neq 0$ is a singular point, i.e., there is another filter vector $\mathbf{w}'$ such that $f_{\mathbf{w}} = f_{\mathbf{w}'}$. By Corollary C.2, $\mathbf{w}'$ differs from $\mathbf{w}$ by more than just layerwise scalings by constants. The monomials of maximal degree appearing in $f_{\mathbf{w}}(x)$ are

$$w_L \star_{s_L} \sigma_r \left( \cdots \star_{s_2} \sigma_r (w_1 \star_{s_1} x) \right) = w_L' \star_{s_L} \sigma_r \left( \cdots \star_{s_2} \sigma_r (w_1' \star_{s_1} x) \right). \tag{56}$$

In particular, $\mathbf{w}$ and $\mathbf{w}'$ also give rise to the same monomial CNN. Now, (Shahverdi et al., 2025, Theorem 4.6) states that $\mathbf{w}$ is a proper $\mathbf{A}$-subnetwork for some $\mathbf{A}$, and that the filters in $\mathbf{w}'$ are shifted versions of the filters in $\mathbf{w}$. These shifts need to satisfy $\tilde{t}_i \in \mathbb{Z}$ and $\tilde{t}_L = 0$ according to (Shahverdi et al., 2025, Remark 4.2).

For the converse direction, we consider a proper $\mathbf{A}$-subnetwork $\mathbf{w}$ that satisfies the assumptions of the statement. To this end, define new weights $\mathbf{w}' = (w_1', \ldots, w_L')$ as:

$$w_i' = \begin{cases} (w_i[t_i+1], w_i[t_i+2], \ldots, w_i[k_i], 0, \ldots, 0), & t_i \geq 0, \\ (0, \ldots, 0, w_i[1], w_i[2], \ldots, w_i[k_i+t_i]), & t_i \leq 0. \end{cases} \tag{57}$$

Since each filter $w_i$ is non-zero and $t_j \neq 0$ for some $j$, it holds that $\mathbf{w} \neq \mathbf{w}'$. Since $\tilde{t}_L = 0$, by leveraging on the equivariance property of convolutions, it is an immediate calculation to verify that $f_{\mathbf{w}} = f_{\mathbf{w}'}$, as desired. $\square$

### C.3 PROOF OF PROPOSITION 4.5

*Proof.* Since $\varphi$ if regular, $\dim(\mathrm{im}(\mathbf{J_W}\varphi)^{\perp}) = \dim(\mathcal{V}) - \dim(\mathcal{M}_{\mathbf{k,s,d},\sigma})$ for every $\mathbf{W} \in S$. Since $\varphi$ has finite fibers, equation 8 defines a family of affine subspaces of $\mathcal{V}$, where only a finite number of subspaces is indexed by a given point in $\varphi(S)$. Therefore, $\dim(U_S) \leq \dim(\varphi(S)) + \dim(\mathcal{V}) - \dim(\mathcal{M}_{\mathbf{k,s,d},\sigma})$. Since $S$ is contained in an algebraic variety strictly contained in $\mathcal{W}$, $\dim(\varphi(S)) < \dim(\mathcal{M}_{\mathbf{k,s,d},\sigma})$. We conclude that $\dim(U_S) < \dim(\mathcal{V})$, implying that $U_S$ must have empty interior. $\square$

## D EXPLICIT EXAMPLES

In this section, we discuss explicit examples of both MLPs and CNNs.

### D.1 MLP

In the following example, we compute the singular locus of a shallow MLP with a polynomial activation, and verify that the subnetworks are critically exposed. The computations are performed via `Macaulay2` (Grayson & Stillman) – a symbolic algebra software.

Consider the shallow MLP with architecture $\mathbf{d} = (2, 2, 1)$ and activation $\sigma(x) = x^3 + x^2$, and parametrization

$$\varphi\left(\begin{bmatrix} a & b \\ c & d \end{bmatrix}, [e, f]\right) = (a^3 e + c^3 f) x_1^3 + (3a^2 be + 3c^2 df) x_1^2 x_2$$
$$+ (a^2 e + c^2 f) x_1^2 + (3ab^2 e + 3cd^2 f) x_1 x_2^2 \qquad (58)$$
$$+ (2abe + 2cdf) x_1 x_2 + (b^3 e + d^3 f) x_2^3 + (b^2 e + d^2 f) x_2^2.$$

We identify the output with its coefficient vector in $\mathbb{R}^7$ relative to the lex-ordered basis $(x_1^3, x_1^2 x_2, x_1^2, x_1 x_2^2, x_1 x_2, x_2^3, x_2^2)$, and write

$$\varphi\left(\begin{bmatrix} a & b \\ c & d \end{bmatrix}, [e, f]\right) = t = (t_1, \ldots, t_7), \qquad (59)$$

where $t_i$ denotes the coefficient of the $i$-th basis monomial. Then the Zariski closure $\overline{\mathcal{M}}_{\mathbf{d},\sigma}$ of the neuromanifold $\mathcal{M}_{\mathbf{d},\sigma}$ in $\mathbb{R}^7$ is the hypersurface $F = 0$, where

$$F = 2t_3 t_4^2 - t_2 t_4 t_5 - 6t_2 t_3 t_6 + 9t_1 t_5 t_6 + 2t_2^2 t_7 - 6t_1 t_4 t_7. \qquad (60)$$

The singular locus of $\overline{\mathcal{M}}_{\mathbf{d},\sigma}$ is the subvariety defined by $\nabla F = 0$. After simplifying the equations, this locus is a 3-dimensional variety cut out by the following polynomials:

$$3t_5 t_6 - 2t_4 t_7, \qquad 3t_3 t_6 - t_2 t_7, \qquad t_4 t_5 - 2t_2 t_7, \qquad t_2 t_5 - 6t_1 t_7,$$
$$t_4^2 - 3t_2 t_6, \qquad t_3 t_4 - 3t_1 t_7, \qquad t_2 t_4 - 9t_1 t_6, \qquad 2t_2 t_3 - 3t_1 t_5, \qquad t_2^2 - 3t_1 t_4. \qquad (61)$$

These equations precisely correspond to the subnetworks, that is, to those weight matrices $\mathbf{W} = (W_1, W_2)$ for which either $e = 0$, or $f = 0$, or $a = c$ and $b = d$.

We now show that the parameter points corresponding to these singularities are critically exposed. To this end, note that the Jacobian of the parametrization with respect to $(a, b, c, d, e, f)$ is

$$\mathbf{J_W}\varphi = \begin{bmatrix} 3a^2 e & 0 & 3c^2 f & 0 & a^3 & c^3 \\ 6abe & 3a^2 e & 6cdf & 3c^2 f & 3a^2 b & 3c^2 d \\ 2ae & 0 & 2cf & 0 & a^2 & c^2 \\ 3b^2 e & 6abe & 3d^2 f & 6cdf & 3ab^2 & 3cd^2 \\ 2be & 2ae & 2df & 2cf & 2ab & 2cd \\ 0 & 3b^2 e & 0 & 3d^2 f & b^3 & d^3 \\ 0 & 2be & 0 & 2df & b^2 & d^2 \end{bmatrix}, \qquad (62)$$

and evaluating at points with $f = 0$ gives

$$\begin{bmatrix} 3a^2e & 0 & 0 & 0 & a^3 & c^3 \\ 6abe & 3a^2e & 0 & 0 & 3a^2b & 3c^2d \\ 2ae & 0 & 0 & 0 & a^2 & c^2 \\ 3b^2e & 6abe & 0 & 0 & 3ab^2 & 3cd^2 \\ 2be & 2ae & 0 & 0 & 2ab & 2cd \\ 0 & 3b^2e & 0 & 0 & b^3 & d^3 \\ 0 & 2be & 0 & 0 & b^2 & d^2 \end{bmatrix}. \tag{63}$$

The tangent directions to the subnetwork $f = 0$ are $(a, b, c, d, e)$; the $c, d$ columns vanish here, so the effective Jacobian on the subnetwork is the $7 \times 3$ matrix formed by the $(a, b, e)$-columns, which we denote by $\mathbf{J}^{\mathrm{sub}}_{f=0}\varphi$. The $3 \times 3$ minor using the first three rows equals $3a^6e^2$. Hence, for general $\mathbf{W}$ with $f = 0$ and $a \neq 0$ and $e \neq 0$, we have that

$$\operatorname{rank} \mathbf{J}^{\mathrm{sub}}_{f=0}\varphi = 3, \tag{64}$$

which shows that

$$\dim\{\, u \in \mathbb{R}^7 \mid (\mathbf{J_W}\varphi)^\top \cdot (t - u) = 0 \,\} = 7 - \operatorname{rank} \mathbf{J}^{\mathrm{sub}}_{f=0}\varphi = 4. \tag{65}$$

As a consequence, the right-hand side of equation 8 has dimension $3+4 = 7$, which equals the ambient dimension of $\mathbb{R}^7$. This shows, from a purely-geometrical perspective, that the set of subnetworks is critically exposed.

## D.2   CNN

Here, we illustrate the technical condition in Theorem 4.6 for a minimal example. We consider the shallow CNN

$$[d \quad e] \cdot \sigma \left( \begin{bmatrix} a & b & c & 0 & 0 \\ 0 & 0 & a & b & c \end{bmatrix} x \right) \tag{66}$$

of stride 2, with filters $w_1 = [a \quad b \quad c], w_2 = [d \quad e]$, with input $x = [x_1 \quad x_2 \quad x_3 \quad x_4 \quad x_5]^\top$, and with activation $\sigma(y) = y^2 + y$. Using `Macaulay2`, we can easily check that the network parametrization map $\varphi$ is regular (away from the parameters $w_1 = 0$ or $w_2 = 0$ mapping to the zero function) and almost everywhere injective.

We can obtain many subnetworks following Definition 7, e.g., by setting $a = 0$, or $a = b = 0$, or $e = 0$, or $a = b = e = 0$. Out of those subnetworks, we will see now that only the latter constitutes a singularity.

The subnetwork $a = b = e = 0$ corresponds to the function $x \mapsto d \cdot \sigma(cx_3)$, which can be embedded in two different ways as a subnetwork into the original architecture:

$$[d \quad 0] \cdot \sigma \left( \begin{bmatrix} 0 & 0 & c & 0 & 0 \\ 0 & 0 & 0 & 0 & c \end{bmatrix} x \right) = [0 \quad d] \cdot \sigma \left( \begin{bmatrix} c & 0 & 0 & 0 & 0 \\ 0 & 0 & c & 0 & 0 \end{bmatrix} x \right). \tag{67}$$

These two subnetworks are in the same fiber of the network parametrization map $\varphi$. Since $\varphi$ is regular and generically one-to-one, this means that the (two equivalent) subnetworks yield a singularity on the neuromanifold. This can be also verified via explicit calculations in `Macaulay2`: For that, we first compute a vanishing ideal of the neuromanifold and then check that the function $x \mapsto d \cdot \sigma(cx_3)$ is indeed a singularity by seeing that the Jacobian of the ideal at that function has lower rank than the expected rank.

For each of the other mentioned subnetworks, namely $a = 0$ or $a = b = 0$ or $e = 0$, there is a unique way of embedding the subnetwork into the original architecture (assuming that all unspecified parameters are non-zero). Thus, they do not cause singularities on the neuromanifold. Again, this can be verified in `Macaulay2` by computing that the rank of the Jacobian of the vanishing ideal of the neuromanifold is the expected one.

We can easily check that this behavior is encoded in the combinatorial condition on the $\tilde{t}_i$ given in Theorem 4.6:

|  | $a = 0$ | $a = b = 0$ | $e = 0$ | $a = b = e = 0$ |
| --- | --- | --- | --- | --- |
| $(\tilde{t}_1, \tilde{t}_2)$ | $(1, \frac{1}{2})$ | $(2, 1)$ | $(0, -1)$ | $(2, 0)$ |

The general intuition is that, for a subnetwork to be embeddable in several ways into a given architecture, the added zeroes in one layer have to be "canceled out" in the next layer (compatible with the stride).

