# OpenReview forum: "Learning on a Razor’s Edge: Identifiability and Singularity of Polynomial Neural Networks"
_ICLR.cc/2026/Conference — ICLR 2026 Poster_

### Official Review · Reviewer_RBeS · 2025-10-23

**Soundness:** 3
**Presentation:** 3
**Contribution:** 3
**Rating:** 4
**Confidence:** 1

**Summary:**

The paper studies the function spaces corresponding to CNNs and MLPs with polynomial activations using tools from algebraic geometry. Due to polynomial activations, these function spaces are manifolds. The authors show that almost everywhere, each function on this manifold corresponds to at-most finitely many parameter values of the network (the identifiability problem) and stronger result for CNNs (almost everywhere uniformly identifiable). The authors also characterize the singularities of the neuromanifolds.


I want to note that I have no training in algebraic geometry and most of the results and ideas in this paper appeared opaque to me. I have set my confidence level to be low to reflect this fact.

**Strengths:**

The paper tackles the important question about the class of functions represented by neural networks. While standard results with non-linear non-polynomial activation functions consider approximation properties, this considers a different of view: that of characterizing the set of functions exactly represented by neural network families. The paper answers some deep questions about this function space such as identifiability and singularity.

**Weaknesses:**

- The authors claim that the identifiability results were previously only known for sigmoid and tanh activation functions. Given that they are more widespread in ML than polynomial activations, I would say that the current results are not as relevant to the community.

- It does not directly give us relevant insights into practically relevant activation functions such as ReLU or sigmoid.

**Questions:**

- Theorem 4.1 requires $r$ to be large enough. Is there a quantitative estimate about how large it should be?

- In Theorem 4.2 it is not clear what "enough coefficients of \sigma are non-vanishing" means.

- The term "generic polynomial" is used throughout the paper, but I could not gather the meaning of this term. This could be because I am not trained in algebraic geometry.

---

> ### Author Response · Authors · 2025-11-16
>
> We thank the reviewer for the valuable feedback.
>
> We wish to comment on the point raised about the relevance of our results **beyond polynomial activations**. To this end, as mentioned in Sec. 5, we wish to highlight that polynomials can (locally) approximate arbitrary continuous functions. This is a core motivation behind neuroalgebraic geometry (see [1], Section 5), whose research program proposes to analyze models with polynomial activations, to which methods from algebra apply, and to then potentially extend the results beyond polynomials. In fact, in the new re-uploaded version of the paper, we have expanded Remark 4.1 with a sketch of an informal argument for extending our main results (for both singularity and identifiability) to "almost all" activations belonging to a given space of smooth functions. However, we leave a complete rigorous proof for future investigation. Moreover, a caveat is that the notion of "almost all" depends on the function space chosen, and it is challenging to describe explicitly. Thus, it is unfeasible to check whether the result applies to a specific activation function. Yet, we see this as a showcase for how results for polynomials can be generalized far beyond the algebraic realm.
>
> Regarding the questions raised in the review:
> - **Degree bounds**: It is actually possible to provide explicit bounds for the degree of the activation function. The proof of Theorem 4.1 relies on the main result from [2], which provides explicit bounds. By using the latter, and by following the proof of Theorem 4.1, it is possible to derive the bound $(6m)^{2(L-1)^{L-1}}$, where $m= 2\max\\{d_1,\ldots,d_{L-1}\\}$. This bound is large, and likely to be very loose. We see our result as the first theoretical proof of the existence of a large enough degree. Moreover, the existence of any bound suffices for polynomial approximation purposes, since it is possible to approximate (continuous) functions with polynomials of arbitrarily-large degree (see Remark 4.1). We agree that including this explicit bound would be helpful. We have incorporated it in the new version of the manuscript (see lines 277-281), which we have re-uploaded.
> - **Clarification of Theorem 4.2**: In the statement of the theorem, we mean that a large enough number of coefficients of the polynomial $\sigma$ must be different from $0$. From the proof it actually follows that it suffices to have a number of non-vanishing coefficients larger than the dimension of the neuromanifold, or, more roughly, than the number of parameters. We have clarified the statement of the theorem in the new version.
> - **Notion of genericity**: By "generic polynomial" we mean a polynomial of a given degree for which the stated property holds for almost all choices of coefficients, except possibly for a set of measure zero. This means that our results hold for "almost all polynomials" (of a given degree). Actually, due to the algebraic nature of the results, this zero-measure set is algebraic, i.e., it is the space of solutions to a system of polynomial equations. In the original version, we explained the notion of generic in the footnote of page 4, which we acknowledge was unclear. In the new version, we have added a background section (Sec. 3.1) where, following feedback by other reviewers, we provide a friendly introduction to the Zariski topology, including the notion of genericity.
>
>
> [1] Marchetti et al., Algebra Unveils Deep Learning -- An Invitation to Neuroalgebraic Geometry, ICML 2025.
>
> [2] Finkel et al., Activation degree thresholds and expressiveness of polynomial neural networks, 2024.

---

> > ### Comment · Reviewer_RBeS · 2025-11-22
> > **Acknowledgement of the Response.**
> >
> > Thank you very much for the response. This resolves many of the technical questions I had. As I have acknowledged in my review, I have very little expertise in this field and have communicated this to the AC. My rating of 4 was given with abundance of caution. I am happy to reconsider my score once you answer these follow-up questions:
> >
> > In Theorem 4.1, it is stated that sigma should be a `generic' polynomial with large enough degree (the bound on the degree is very large, but I agree that it is likely a sub-optimal bound). Do the following hold:
> > 1. Any arbitrary continuous function on a compact set on R can be approximated by a generic polynomial ? (I believe this should be true)
> > 2. Can such a generic polynomial be made "regular" while also approximating a function such as a sigmoid ? i.e, can it have O(1) bounded derivatives over a compact set (say [-1,1]). I ask this since uniform convergence does not capture the notion of regularity, but this can be important for optimization.
> > 3. Can 1. and 2. hold even with the non-vanishing coefficient assumption in Theorem 4.2?

---

> > > ### Author Response · Authors · 2025-11-22
> > >
> > > We thank the reviewer for the follow-up questions, which we address below.
> > >
> > > 1. Yes. By the Stone-Weierstrass approximation theorem, any continuous function on a compact set can be approximated (in the uniform norm) by a polynomial $q$. Since the genericity condition in Theorem 4.1 is dense in the coefficient space of $q$ (i.e., it holds for almost all polynomials), we can perturb the coefficients of $q$ slightly, obtaining a new polynomial $p$ which is also uniformly approximating, and that satisfies the condition from Theorem 4.1.
> > >
> > > 2. We acknowledge that regularity is an important point, and we agree that uniform approximation is not, by itself, sufficient to this end. Yet, polynomials still enjoy the property the reviewer asks about. Specifically, given a positive integer $k$, the Stone-Weierstrass theorem can be refined to provide polynomial approximations of a function of class $C^k$ in the uniform norm of its derivatives up to $k$-th order **simultaneously** – see, e.g., Theorem 2.12 in [1]. This means that not only the polynomial will approximate the given function, but its derivatives will also approximate the derivatives of that function. Moreover, the argument from point 1) still applies verbatim to this context, meaning that this stronger approximation property holds under the genericity condition of Theorem 4.1
> > >
> > > 3. Yes. This follows from the fact that the hypothesis of Theorem 4.2 (non-vanishing of coefficients) is, in fact, a generic condition. In particular, it is satisfied by almost all polynomials (of a given large-enough degree) or, more precisely, by polynomials whose coefficients lie outside of an algebraic set in the coefficient space (i.e., the set where some coefficients vanish). Thus, the arguments from points 1) and 2) above apply to the condition from Theorem 4.2 as well.
> > >
> > > We hope that these points clarify some of the details behind the extension of our results via polynomial approximations.
> > >
> > > [1] D. Pérez and Y. Quintana, A survey on the Weierstrass approximation theorem, Divulgaciones Matematicas, 2008.

---

> > > > ### Comment · Reviewer_RBeS · 2025-11-23
> > > > **Response**
> > > >
> > > > Thanks for the clarification. I now see that "density" in the Zariski topology behaves similar to density in Euclidean spaces, atleast in the context of polynomial approximation. I will increase my score.

---

### Official Review · Reviewer_jF6i · 2025-10-29

**Soundness:** 3
**Presentation:** 3
**Contribution:** 3
**Rating:** 8
**Confidence:** 3

**Summary:**

The paper studies the (dis)connectedness and the existence of singularities in the parameter space of ReLU neural networks whose architectures admit a DAG computational graph representation. The authors then leverage these results to study whether these elements can exert impact on the dynamics of gradient flow (GF) of standard training algorithms.

**Strengths:**

I generally like the paper. Here are several remarkable points:
1. The conservative law for ReLU networks in the DAG case is very elegant.
2. Theorem 1 makes a very nice connection to flow problem in graph theory.
3. Theorem 2 - Proposition 5 to 7 provide a very clean picture on the dynamics of gradient flow in the existence of singularities

**Weaknesses:**

1. I have a hard time distinguishing what are the main contributions and what are already proved in the literature. Authors might want to re-organize the section 2, and credit properly all the results (theorems, propositions, definitions) if they are ever taken/inspired by previous works.

2. Do the author forget to define the notion of stable by forward/backward edges in the announcement of Theorem 1? Otherwise, I believe that Theorem 1 needs rephrasing to be easier to understand.

**Questions:**

1. Do Proposition 6 - 7 imply that singularities are truly rare? It seems to me that the limit of GF can still be a singularity (or a sparse subnetwork). If the GF dynamics does not bias towards sparse subnetworks, do you have any idea which points are preferable for the convergence of GF?

---

> ### Comment · Reviewer_jF6i · 2025-11-13
> **Updated review**
>
> I updated the review, since the old one corresponds to another paper. Sorry the authors for the inconvenience

---

> ### Author Response · Authors · 2025-11-16
>
> We thank the reviewer for the comments and the feedback.
>
> We wish to address the weakness raised:
> 1. We agree that polynomial activations are not too popular in practice. Yet, we wish to stress that polynomials can (locally) approximate arbitrary continuous functions, including ReLU. This is a core motivation behind neuroalgebraic geometry (see [1], Section 5), whose research program proposes to analyze models with polynomial activations, to which methods from algebra apply, and to then potentially extend the results beyond polynomials. In fact, in the new re-uploaded version of the paper, we have expanded Remark 4.1 with a sketch of an informal argument for extending our main results (for both singularity and identifiability) to "almost all" activations belonging to a given space of smooth functions. However, we leave a complete rigorous proof for future investigation. Moreover, a caveat is that the notion of `almost all’ depends on the function space chosen, and it is challenging to describe explicitly. Thus, it is unfeasible to check whether the result applies to a specific activation function (e.g., ReLU). Yet, we see this as a showcase for how results for polynomials can be generalized far beyond the algebraic realm.
> 2. We acknowledge that the bound from Theorem 4.1 is extremely large (note that we have made this bound explicit in the re-uploaded version of the manuscript, as suggested by other reviewers). However, we wish to stress that several commonly-deployed activation functions — such as Tanh, Sigmoid, and smooth variants of ReLU such as GELU  – are analytic functions, i.e., they admit a Taylor expansion. Thus, these activations can be written as power series or, roughly speaking, as "polynomials with infinite degree". Thus, even though the bounds from our results are large, we believe they are likely to reflect the behavior of actually deployed models. However, making this argument via "infinite-degree polynomials" rigorous is challenging, and essentially equivalent to the polynomial approximation approach mentioned in the previous point, and in Remark 4.1.
> 3. The reviewer is correct; being critically exposed for a set of parameters $S$ means that it will contain critical points for a positive-measure set of $u$’s (i.e., of data). However, we wish to highlight that actually, we show that when $S$ is the set of subnetworks, this holds for **almost all** $u$. In other words, this will happen with probability equal to $1$ (assuming the probability distribution is absolutely continuous, technically speaking). This is because we work in the Zariski topology, so the fact that $U_S$ is open implies that it is dense. This density phenomenon is a major advantage of the algebraic setting. We acknowledge that this was not clear in the original version of the manuscript. In the new version, we have remarked this in Section 4.3 (see lines 364-367). Moreover, following suggestions by other reviewers, we have added a background section (Sec. 3.1) discussing the Zariski topology and its benefits in a friendly manner.

---

> ### Author Response · Authors · 2025-11-16
>
> Lastly, we wish to answer the question raised in the review.  We acknowledge that Section 3.3 (numbering from the new version) is rather short. Our intention was to briefly summarize the optimization aspect, which is discussed in detail in the works cited in that section. Nonetheless, we wish to highlight that in terms of the ideal goal of optimization, parameter space and function space are equivalent, since one is mapped (surjectively) into the other. More precisely, denote by $\mathcal{M}$ the neuromanifold and consider any loss function $\mathcal{L} \colon \mathcal{M} \rightarrow \mathbb{R}$ over it, which induces a loss function $\mathcal{L} \circ \varphi \colon \mathcal{W} \rightarrow \mathbb{R}$ over the parameter space $\mathcal{W}$. Then a parameter $W  \in \mathcal{W}$ is a global minimum of the latter if and only if the corresponding function $\varphi(W)$ is a global minimum of $\mathcal{L}$ over $\mathcal{M}$.
>
> Now, when $\mathcal{L}$ is MSE (Eq. 4), it can be rewritten precisely as a quadric $Q(\bullet - u)$, for suitable quadratic form $Q$ and vector $u$ depending on data $\mathcal{D}$. We did not include the precise expressions for $Q$ and $u$ for conciseness; they can be found, e.g., in [2]. We have remarked this in the new version.
>
> Lastly, we remark that in practice, optimization in parameter space is not equivalent to function space, since it is usually performed via (variations of) gradient descent. The gradient flow of $\mathcal{L}$ over $\mathcal{W}$ and the one of $\mathcal{L} \circ \varphi$ over $\mathcal{M}$ might exhibit different dynamics (e.g., different local optima, critical points, attractor basins, etc.). However, we point out that in the definition of critical exposedness, we consider critical points of $\mathcal{L} \circ \varphi$ over the parameter space $\mathcal{W}$, where gradient descent is performed in practice. Thus, we believe that this definition, and the resulting theory, is faithful to practice. The reason we mention optimization in function space in Sec. 3.3 is to enable a geometric interpretation of critical exposedness (see Eq. 8, and the surrounding discussion).
>
> [1] Marchetti et al., Algebra Unveils Deep Learning -- An Invitation to Neuroalgebraic Geometry, ICML 2025.
>
> [2] Trager et al., Pure and Spurious Critical Points: A Geometric Study of Linear Networks, ICLR 2020.

---

> ### Comment · Reviewer_jF6i · 2025-11-17
>
> Thanks the authors for the responses. I keep my score as it is.

---

### Official Review · Reviewer_WScj · 2025-10-31

**Soundness:** 3
**Presentation:** 3
**Contribution:** 4
**Rating:** 6
**Confidence:** 4

**Summary:**

The paper investigates the geometry of neural networks with polynomial activation functions through algebraic geometric tools. The authors claim three main contributions:
Identifiability results: For MLPs with generic polynomial activations, almost all functions have finitely many parameter representations; for CNNs, the parametrization is generically one-to-one.
Singularity characterization: Sparse subnetworks constitute singular points of neuromanifolds for both MLPs and CNNs.
Critical exposedness: Subnetworks of MLPs are "critically exposed" (contain critical points of the loss with positive probability), providing a geometric explanation for sparsity bias. CNNs do not exhibit this property.

**Strengths:**

Important theoretical questions: The paper addresses fundamental issues in deep learning theory - identifiability, singularities, and sparsity bias - that have significant implications for understanding optimization and generalization.
Novel geometric perspective: Connecting sparsity bias to singularities of neuromanifolds is creative and could provide new insights into the lottery ticket hypothesis.
Architectural comparison: The distinction between MLP and CNN geometry regarding critical exposedness is novel and aligns with empirical observations about their different behaviors.

**Weaknesses:**

The paper relies heavily on citations in a way that makes the intuition of the proofs difficult to follow . I would appreciate a more self contained mathematical exposition. I would be happy to improve my rating if more exposition was provided, as well as the following are addressed.

Issues with specific proofs:
Theorem 4.1 (MLP Identifiability)
- The constraint β₁ > 6m² - 6m appears without much justification. Why this specific bound?
- The dependence on this proof and a good bit of discussion in the paper on the Zariski topology necessitates a more thorough exposition/ explanation of this topological space, as it is a rather different topology than commonly used in Learning.

Theorem 4.2 (MLP Singularities)
- There should be more said about the dominance of σ ◦ fW′  . In particular it isnt clear that the image of σ ◦fW′ having a non-empty interior is sufficient to show Zariski density .

Theorem 4.6 (CNN Singularities)
More said at all points.

**Questions:**

Add concrete examples: Provide explicit small examples (e.g., 2-layer networks) where singularities and exposedness can be computed directly, if possible.

---

> ### Author Response · Authors · 2025-11-16
>
> We thank the reviewer for the thorough feedback and the encouraging comments.
>
> We acknowledge that our paper can greatly benefit from more self-containment, and clearer explanations, especially regarding some concepts from algebraic geometry. We have uploaded a new version of the manuscript, where we have improved the exposition in several points.
>
> Regarding the specific points raised in the weakness section:
> - **Constraint** on $\beta_1$: The constraint mentioned in the review comes directly from [1], which studies identifiability of MLPs with monomial activations ($\sigma(x) = x^r$). As explained in the proof summary in the main body of the paper, our strategy to prove the polynomial case is to reduce to the monomial case via a subtle decomposition of the MLP, and then to rely on the results from [1]. Thus, it is necessary to choose degrees that respect the bounds from [1]. In the new version, for the sake of self-containment, we have included the statement of the main result from [1] in the appendix (Theorem B.3). where the bound is stated explicitly. This statement is referred to in the proof of Theorem 4.1.
> - **Zariski topology**: We agree that an explanation of the Zariski topology would improve the exposition, since it is important in several points in the paper. In the new version, we have added a background section (Sec. 3.3), where we overview the basic notions from Zariski topology, as well as its main benefits, which are leveraged upon throughout the work.
> - **Dominance in Theorem 4.2**: We agree that this point is unclear. This step of the proof relies on a fundamental property of the Zariski topology: any non-empty open set is dense in this topology. In particular, any set with a non-empty interior is dense. This property is a fundamental tool in algebraic geometry. As such, we have discussed it in the afore-mentioned Sec. 3.3, and clarified the proof of Theorem 4.2.
>
> Unfortunately, we did not understand the weakness about Theorem 4.6 (CNN Singularities). May we ask the reviewer to expand on it? We would be happy to address it.
>
> Regarding the question raised in the review, we agree that **examples** help clarifying the statements about singularity and exposedness. An illustrative example is already provided in Sec. 4.2 by linear networks (i.e., $\sigma(x) = x$) with a bottleneck, where the neuromanifold corresponds to low-rank matrices, whose singular points are known, and can be identified with subnetworks. Moreover, following the reviewer’s suggestion, we have added a new concrete example (Sec. D in the appendix), including detailed computations. We consider a small network with activation $\sigma(x) = x^3 + x^2$, and explicitly compute, via symbolic algebra software, the singular locus and exposedness. In particular, we find out that all singular points correspond to subnetworks, corroborating the conjecture mentioned in Sec. 5. This example is referenced in the main body (Sec. 4.2).
>
> [1] Finkel et al., Activation degree thresholds and expressiveness of polynomial neural networks, 2024.

---

> > ### Comment · Reviewer_WScj · 2025-11-26
> >
> > I thank the reviewers for these clarifications and changes, which mostly address my concerns.

---

> > > ### Author Response · Authors · 2025-11-28
> > >
> > > We thank the reviewer for the follow-up. Earlier, you mentioned that additional exposition would be helpful, so please let us know if there is anything further you feel would improve your assessment. We would be happy to clarify.

---

### Official Review · Reviewer_ZhpW · 2025-10-31

**Soundness:** 4
**Presentation:** 3
**Contribution:** 4
**Rating:** 8
**Confidence:** 4

**Summary:**

The paper studies the function spaces (“neuromanifolds”) of deep MLPs and CNNs with polynomial activations, using tools from algebraic geometry. The core results:

Identifiability. For MLPs with a generic sufficiently high-degree polynomial activation, the parametrization is generically finite-to-one; hence the dimension of the neuromanifold equals the number of parameters (Theorem 4.1). For CNNs, it is generically one-to-one and regular off the zero fiber (Theorem 4.4).

Singularities. Subnetworks (deactivating neurons/filters) yield singular points: fully characterized for CNNs (Theorem 4.6) and partially for MLPs (Theorem 4.2).

Optimization bias. The paper introduces “critically exposed” parameter sets. Strict subnetworks of MLPs are critically exposed for quadratic losses (Theorem 4.3), but not for CNNs (Proposition 4.5). This gives a geometric account of sparsity bias in MLPs.

**Strengths:**

Conceptual advancement. Provides a clean algebraic–geometric framework for identifiability and singularity in deep networks with polynomial activations, extending prior results beyond monomials.

Generality of MLP result. Finite identifiability for generic polynomials closes a dimension conjecture (dimension = #params) and generalizes known tanh/sigmoid cases to large-degree polynomials (Theorem 4.1).

CNN result. The regularity and generic injectivity of CNN parametrizations off the zero fiber (Theorem 4.4) is technically strong and explains why CNN singularities are mild and do not create optimization equilibria.

Sparsity bias lens. The critical exposedness notion and proofs (Theorem 4.3 vs Prop. 4.5) give a principled account of why MLPs tend to collapse to sparse subnetworks whereas CNNs typically recover from near-zero initializations—matching known empirical phenomena.

**Weaknesses:**

Activation assumptions: Many results require “generic” high-degree polynomials (often with σ(0)=0 and nonzero top coefficients). Practical popular activations (ReLU, GELU, tanh) are non-polynomial; while the authors argue approximation plausibility (Remark 4.1, §5), formal transfer to non-polynomial nets is not proven.

Optimization link: While “critically exposed” is compelling and geometric, the paper doesn’t classify whether exposed subnetworks are local minima vs saddles, which matters for SGD dynamics and generalization.

**Questions:**

1) Can you sketch the proof of  the limit argument suggested in Remark 4.1: under what conditions do MLP/CNN singularities persist under uniform polynomial approximation of non-polynomial activations?

2) Are there settings where subnetwork critical points in MLPs are provably local minima with non-negligible measure?


3) Can dimension counts for U_S (Eq. 8) yield measure bounds for how often subnetworks are equilibria?

---

> ### Author Response · Authors · 2025-11-16
>
> We thank the reviewer for the comments, the feedback, and the appreciation.
>
> We wish to answer the questions raised:
>
> 1. We believe that the **polynomial approximation** approach is an important point. In the new re-uploaded version of the paper, we have expanded Remark 4.1 with a sketch of an informal argument for extending our results (on both singularity and identifiability) to "almost all" activations belonging to a given space of smooth functions. However, we leave a complete rigorous proof for future investigation, since it would probably require subtle details from functional analysis. Moreover, a caveat is that the notion of "almost all" depends on the function space chosen, and it is challenging to describe explicitly. Yet, we see this as a showcase for how results for polynomials can be generalized far beyond the algebraic realm.
> 2. We agree that determining the **type of critical points** (especially, local minima) is a fundamental problem, as we mentioned in Sec. 5. However, we believe that this a challenging task that would involve careful analysis of Hessians, and we are unaware of examples where subnetworks are provably local minima for a positive-measure set of $u$’s. Thus, we see this as a major future direction. Moreover, we wish to mention the following interesting phenomenon. When noise is taken into account in the dynamics, arbitrary critical points can become (local) attractors. This is a general principle from stochastic processes, and has been applied to deep learning in [1]. The latter shows that for stochastic gradient descent, with some assumptions on the type of noise, a critical point of the loss can attract the stochastic dynamics, in a probabilistic sense, if the noise is strong enough (see Theorem 4.1 and 4.2 in [1]). We believe that this showcases that critical points of any type play an important role in terms of convergence of the learning dynamics, and corroborates the significance of the notion of critical exposedness. We have incorporated this discussion in the re-uploaded version of the manuscript (see end of Sec. 3.3).
> 3. Regarding the **dimension** of $U_S$, we wish to highlight that the notion of critically exposedness requires $U_S$ to have non-empty interior, implying that its dimension will coincide with the dimension of the ambient space $\mathcal{V}$. Thus, in our view, the dimension of $U_S$ is not an interesting quantity for critically exposed sets $S$. Even further, when $S$ is the set of subnetworks, the proof of Theorem 4.3 shows that $U_S$ is full-measure in $\mathcal{V}$. Thus, $S$ will contain critical points with probability $1$ with respect to $u$ (according to any absolutely continuous probability measure, technically speaking). This follows from the fact that we work with the Zariski topology. We acknowledge that this was unclear in the original version of the manuscript, and we have remarked on it in the new version (see lines 364-367).
>
>
> [1] Chen et al., Stochastic Collapse: How Gradient Noise Attracts SGD Dynamics Towards Simpler Subnetworks, NeurIPS 2023.

---

> > ### Comment · Reviewer_ZhpW · 2025-11-17
> >
> > Thank you for your response.
> >
> > Another question that I have is that your results state (correct me if I'm wrong) that there are only finitely many parameter configurations that can reproduce a given function encoded by a neural network. This seems to go against the current theories for explaining generalisation in neural networks, which posit that many different optima actually encode the same function, and that would be why ending up on different optima still yields the same performance. This also contradicts the sort of low-rank/neural collapse/simplicity arguments which state that many parameter directions are redundant and thereby the typical functions that are learnt are actually much simpler than what could be have been attained with the full expressive power, and in particular, these typical functions can be reproduced by many different parameter configurations (exploiting symmetries etc). Your results suggest otherwise, so how do you think about this apparent tension with other theories ?

---

> > > ### Author Response · Authors · 2025-11-18
> > >
> > > We thank the reviewer for the response, and for the further question, which we find interesting. We wish to highlight that Theorem 4.1 is concerned with **generic** fibers: it shows that, for almost all functions parametrized by the model, there are only finitely-many parameters corresponding to them. However, some special functions have larger fibers. In fact, subnetworks have **infinite** fibers/symmetries, due to their degenerate nature. We discuss this rigorously in lines 240-246 (numbering from the re-uploaded version). This property shows that subnetworks are extremely special parameters, and is crucial in the proof of their singularity (Theorem 4.2).
> > >
> > > In our view, this aligns with the theories mentioned by the reviewer. When the model learns a subnetwork – which, as we argue via critical exposedness, is an expected behavior – there is a large degree of redundancy in the paremetrization of the learnt function. Moreover, the degree of redundancy (i.e., the dimension of the infinite fiber) is controlled by the sparsity/simplicity of the learnt function: smaller/sparser subnetworks correspond to larger fibers (see line 246). Indeed, from a broader perspective, we see our work as fitting into the line of literature about the implicit biases of neural networks such as low rank, simplicity, and neural collapse. As we briefly discuss in the introduction (lines 89-95), we believe that our results provide a geometric perspective on these biases, explaining the special role of subnetworks (i.e., of sparse parameters) in the function space parametrized by the model (see Figure 1 for an intuitive illustration).

---

### Official Review · Reviewer_nwJE · 2025-11-01

**Soundness:** 3
**Presentation:** 3
**Contribution:** 3
**Rating:** 8
**Confidence:** 3

**Summary:**

The paper studies the function spaces (neuromanifolds) parameterized by deep MLPs and CNNs with generic high-degree polynomial activations. It proves finite identifiability for MLPs (generic outputs come from finitely many parameters) and generic one-to-one identifiability for CNNs, hence the dimension equals the number of parameters in both settings. It then characterizes singular points of these neuromanifolds: (i) for MLPs, many subnetworks yield singularities and are shown to be critically exposed (they occur as critical points of squared-loss for a set of targets with nonempty interior); (ii) for CNNs, all singularities are precisely the subnetworks with edge zero-padding that satisfy an integrality constraint, and such sets are not critically exposed (away from the zero fiber). This geometric picture explains sparsity bias in MLPs but not in single-channel CNNs and resolves a long-standing dimension conjecture from prior work on polynomial networks. Figures 1--2 visualize how subnetworks create singular points and how MLP cuspidal-type vs. CNN nodal singularities differ.

**Strengths:**

* Originality: Moves from monomial/linear models to generic polynomial activations; formalizes critical exposure; delivers complete CNN singularity characterization.
* Quality: Careful use of fiber-dimension, Vandermonde invertibility, and toric lattice ideals; clean separation between parametrization criticality and image singularity (Appendix A). Proof architecture is transparent via modular lemmas.
* Clarity: Clear definitions of neuromanifolds/subnetworks; optimization setup with quadratic loss; intuitive figures (Fig. 1--2) and didactic examples (nodal vs. cuspidal curves in Fig. 3) to illustrate singular behaviors.
* Significance: Fixes dimension = parameter count generically for both MLPs and CNNs; connects geometry to sparsity bias and to the presence/absence of spurious critical points across architectures.

**Weaknesses:**

* Generality vs. practicality: Many results require very high polynomial degree and generic coefficients. Concrete degree thresholds are not explicit beyond \(r \gg 0\) (depends on architecture). Giving quantitative bounds (even conservative ones) would improve applicability.
* Model scope: CNN analysis is single-channel, 1D; multi-channel and higher-D details are asserted “similarly” but not proved. Because modern CNNs are multi-channel, a pathway or obstacles to generalization would be valuable.
* Singularity coverage in MLPs: The paper shows many singularities arise from subnetworks but leaves open whether all singularities do (contrast with linear MLPs and with CNNs, where a full characterization is given). Clarifying non-subnetwork singularities would strengthen the picture.
* Type of critical points: Criticality vs. local minima vs. saddles is not analyzed; given the optimization motivation (sparsity bias), even partial results or conjectures on stability types would be informative.
* Beyond polynomial activations: While approximation arguments are discussed (Remark 4.1), formal extension to ReLU/Tanh/Softmax is left for future work; clarifying which parts port over under approximation limits would broaden impact.

**Questions:**

Non-polynomial activations: Can the polynomial approximation idea in Remark 4.1 be made quantitative (e.g., stability of singularity types under uniform approximation on compact sets)? Which parts of identifiability/exposedness survive in the ReLU or tanh settings?

---

> ### Author Response · Authors · 2025-11-16
>
> We thank the reviewer for the extensive and detailed feedback, as well as the encouraging comments regarding our results. We wish to address the weaknesses raised in the review.
>
> - **Degree bounds**: It is actually possible to provide explicit bounds for the degree of the activation function. The proof of Theorem 4.1 relies on the main result from [1], which provides explicit bounds. By using the latter, and by following the proof of Theorem 4.1, it is possible to derive the bound $(6m)^{2(L-1)^{L-1}}$, where $m= 2\max\\{d_1,\ldots,d_{L-1}\\}$. As mentioned by the reviewer, this bound is extremely conservative, and likely to be very loose. We see our result as the first theoretical proof of the existence of a large enough degree. Moreover, the existence of any bound suffices for polynomial approximation purposes, since it is possible to approximate (continuous) functions with polynomials of arbitrarily-large degree (see Remark 4.1).  We agree that including this explicit bound would be helpful. We have incorporated it in the new version of the manuscript (after the statement of the theorem), which we have re-uploaded.
> - **Higher-dimensional and multi-channel CNNs**: We acknowledge that our CNN results are stated for 1D single-channel convolutions. Regarding the 1D aspect, our results actually extend to higher dimensions. Indeed, all the results from [2] (on which our arguments are based upon) hold verbatim, with the same proofs, for convolutions in any dimension. They are stated for 1D convolutions for simplicity; higher dimensions require multi-indices (both for the input and the filters), making the formulas hard to read (e.g., Equation 10). We have remarked this in the new version (see lines 440-442). Instead, the multi-channel case is more subtle. Indeed, additional symmetries arise in the fibers due to permutations of channels. Extending the results from [2] to multiple channels is significantly challenging. We believe that it goes beyond the scope of this work, but we acknowledge its importance and plan to consider this direction in future work.
> - **Singularity coverage**: We agree that a complete characterization of singular points for MLPs is important. In this work, we were able to show that for CNNs, all singularities indeed arise from subnetworks. However, for MLPs, as stated in the limitations (Sec. 5), it remains open to prove that all singularities originate from subnetworks. An illustrative example in that direction is the case of linear networks ($\sigma(x) =x$), discussed in the second paragraph of Sec. 4.2, where subnetworks can be explicitly seen to exhaust all the singularities. Moreover, following feedback by other reviewers, we have added an explicit example of a small network (see Sec. D), where we manually check that all singularities correspond to subnetworks. We strongly believe that this property holds for general MLPs, and consider this an important but challenging conjecture for future work.
> - **Type of critical points**: We believe that determining the type of critical points (especially, local minima) is a fundamental problem, as mentioned in Sec. 5. We do not have formal conjectures in this direction at the moment, since the type of critical point can behave non-uniformly with respect to $u$. However, we wish to mention the following interesting phenomenon. When noise is taken into account in the dynamics, arbitrary critical points can become (local) attractors. This is a general principle from stochastic processes, and has been applied to deep learning in [3]. The latter shows that for stochastic gradient descent, with some assumptions on the type of noise, a critical point of the loss can attract the stochastic dynamics, in a probabilistic sense, if the noise is strong enough (see Theorem 4.1 and 4.2 in [3]). We believe that this showcases that critical points of any type play an important role in terms of convergence of the learning dynamics, and corroborates the significance of the notion of critical exposedness. We have incorporated this discussion in the new version (see end of Sec. 3.3).

---

> ### Author Response · Authors · 2025-11-16
>
> Lastly, we wish to answer the question about **polynomial approximation**. We indeed believe that our results extend beyond polynomials. In the new re-uploaded version of the paper, we have expanded Remark 4.1 with a sketch of an informal argument for extending our results (on both singularity and identifiability) to "almost all" activations belonging to a given space of smooth functions. However, we leave a complete rigorous proof for future investigation. Moreover, a caveat is that the notion of "almost all" depends on the function space chosen, and it is challenging to describe explicitly. Thus, it is, unfortunately, unfeasible to check whether the result, in the current formulation, applies to a specific activation function, such as ReLU and Tanh. Similarly, we believe that it is extremely challenging to determine what type of singularities persist at the limit; while interesting, this problem is probably a deep question in singularity theory, to which we do not have a concrete answer at the moment, but that we plan to explore in the future.
>
> [1] Finkel et al., Activation degree thresholds and expressiveness of polynomial neural networks, 2024.
>
> [2] Shahverdi et al., On the Geometry and Optimization of Polynomial Convolutional Networks, AISTATS 2025.
>
> [3] Chen et al., Stochastic Collapse: How Gradient Noise Attracts SGD Dynamics Towards Simpler Subnetworks, NeurIPS 2023.

---

### Author Response · Authors · 2025-11-16
**General Message to all Reviewers**

We thank all the reviewers for their comments. Following feedback, we have uploaded a new version of the manuscript, where we have improved the exposition, and incorporated new discussions. The new text is **highlighted in blue**. The main changes are:

- **Polynomial approximation**: Several reviewers asked about the extension of our result beyond polynomials, which we had hinted towards in the original version. We have extended Remark 4.1 by discussing an informal sketch for an argument to extend our results to "almost all" smooth functions (in an appropriate sense, depending on the topology of the chosen function space) via polynomial approximation. However, we believe that a full formal argument would require plenty of details from functional analysis, probably resulting in a whole work on its own.
- **Explicit bounds**: We have included explicit bounds for the degree of $\sigma$ in Theorem 4.1 and Theorem 4.3.
- **Type of critical points**: We have included a paragraph in Sec. 3.3 discussing the fact that our work focuses only on critical points, and motivating this from the perspective of stochastic dynamics.
- **Exposition**: We have added a new section (Sec. 3.1) containing a friendly introduction to the Zariski topology and the notion of genericity. This section is referred to in several points throughout the paper, where the special properties of the Zariski topology are leveraged upon.
- **Example**: We have added an example in the appendix (Sec. D) of a small network where we manually compute singularities and exposedness, corroborating the conjecture that subnetworks exhaust all the singularities.

We hope that these additions help to address the points raised in the reviews. Below, we reply individually to the reviewers, expanding on the above points.

---

### Meta-Review · Area_Chair_WKZw · 2025-12-27

**Summary:**

The submission develops an algebraic–geometric framework for understanding neuromanifolds induced by deep networks with sufficiently generic polynomial activations, with strong results on both identifiability and singularities. The core technical contributions are (i) generic finite-to-one identifiability for MLPs and generic one-to-one identifiability for CNNs (implying the neuromanifold dimension matches parameter count), and (ii) a subnetwork-driven account of singular point; complete for the CNN setting and substantial (though not fully exhaustive) for MLPs. A key conceptual advance is the notion of critical exposedness, showing strict subnetworks of MLPs arise as critical points for a full-measure set of targets under quadratic loss, while CNN subnetworks are not critically exposed away from the zero fiber, offering a geometric explanation for differing sparsity-bias behavior across architectures.

The general consensus of the reviewers was already quite positive and after the discussion/rebuttal score increases were indicated which I presume due to the special situation did not happen or have been reverted. Without the reviews/rebuttals the paper was already beyond the acceptance threshold with the aforementioned increases even further.

**Reviewer Concerns:**

mostly exposition and one low-confidence-reviewer asked for more details: largely on practicality/generalization beyond polynomials, quantitative thresholds, and self-containment of the exposition; the rebuttal seems to have improved these points by adding explicit (conservative) degree bounds, a better introduction to Zariski genericity/density, clarifications in the dominance arguments, and a concrete small-network example supporting the "singularities come from subnetworks" conjecture in the MLP case.

**Reviewer Scores:**

scores are high already and two reviewers indicated further increase. Acceptance as poster seems obvious - i am not sure though that the pushes would be high enough to merit an oral. but as I am guessing here pretty much as anybody else, I recommended a poster for now but would not mind if my decision is bumped up to oral.

---

### Decision · Program_Chairs · 2026-01-26

Accept (Poster)